# TissueMiner: A multiscale analysis toolkit to quantify how cellular processes create tissue dynamics

Raphaël Etournay[1,2†], Matthias Merkel[3,4†], Marko Popović[3†], Holger Brandl[1†], Natalie A Dye[1], Benoît Aigouy[5], Guillaume Salbreux[3,6], Suzanne Eaton[1*], Frank Jülicher[2*]

[1]Division of Cell Polarity, Max Planck Institute of Molecular Cell Biology and Genetics, Dresden, Germany; [2]Institut Pasteur, Paris, France; [3]Max Planck Institute for the Physics of Complex Systems, Dresden, Germany; [4]Department of Physics, Syracuse University, Syracuse, United States; [5]Institut de Biologie du Développement de Marseille, Marseille, France; [6]The Francis Crick Institute, Lincoln's Inn Fields Laboratories, London, United Kingdom

*For correspondence: eaton@mpi-cbg.de (SE); julicher@pks.mpg.de (FJ)

†These authors contributed equally to this work

**Abstract** Segmentation and tracking of cells in long-term time-lapse experiments has emerged as a powerful method to understand how tissue shape changes emerge from the complex choreography of constituent cells. However, methods to store and interrogate the large datasets produced by these experiments are not widely available. Furthermore, recently developed methods for relating tissue shape changes to cell dynamics have not yet been widely applied by biologists because of their technical complexity. We therefore developed a database format that stores cellular connectivity and geometry information of deforming epithelial tissues, and computational tools to interrogate it and perform multi-scale analysis of morphogenesis. We provide tutorials for this computational framework, called TissueMiner, and demonstrate its capabilities by comparing cell and tissue dynamics in vein and inter-vein subregions of the *Drosophila* pupal wing. These analyses reveal an unexpected role for convergent extension in shaping wing veins.

## Introduction

Understanding how cells collectively shape a tissue is a long-standing question in developmental biology. We recently addressed this question by analyzing morphogenesis of the *Drosophila* pupal wing at cellular resolution (*Etournay et al., 2015*). To understand the cellular contributions to pupal wing shape changes, we quantified the spatial and temporal distribution of both cell state properties (e.g. cell area, shape and packing geometry), as well as dynamic cellular events like rearrangements, divisions, and extrusions. We quantitatively accounted for wing shape changes on the basis of these cellular events. By combining these analyses with mechanical and genetic perturbations, we were able to develop a multiscale physical model for wing morphogenesis and show how the interplay between epithelial stresses and cell dynamics reshapes the pupal wing.

Researchers interested in epithelial dynamics face similar challenges in processing and analyzing time-lapse movie data. Quantifying epithelial dynamics first requires image-processing steps including cell segmentation and tracking, to digitalize the time-lapse information. Recently, software tools for segmentation and tracking have become generally available (*Aigouy et al., 2010*; *Mosaliganti et al., 2012*; *Sagner et al., 2012*; *Barbier et al., 2015*; *Cilla et al., 2015*; *Wiesmann et al., 2015*; *Heller et al., 2016*; *Aigouy et al., 2016*). However, more advanced analysis

**eLife digest** Cells interact, divide, rearrange and change shape to build an organ during development. Modern microscopy and computer technology can follow each individual cell of an entire organ in a living organism. However, to understand how the complex choreography of cells leads to well-shaped organs, researchers need tools to help the store and analyze the large amounts of data generated. Tools are also needed to visualize and quantify the complex cell behaviors in an easy and flexible manner.

During its development, a fruit fly's wings become divided into distinct regions separated by tubular supports called veins. Early on in development, the vein cells are indistinguishable from their neighbors, but at late stages, vein cells become a different shape. Veins also become narrower, which is assumed to be due to the number of vein cells falling. However, the way in which cells behave to bring about these changes has not been studied in detail.

Etournay, Merkel, Popović, Brandl et al. have now developed a toolkit called TissueMiner that enables users to store large amounts of data about cells and analyze how cells collectively shape an organ. TissueMiner was then used to identify vein cells at late stages of wing development and follow them backward in time to reveal their position at early stages. This showed that veins become narrower and more elongated because the cells that make up the veins shrink more than cells in other regions.

TissueMiner was then used to show that vein cells specifically rearrange and elongate to produce thinner regions, while the number of cells increases slightly because the cells divide. These results suggest that the cell behaviors responsible for making veins elongate and narrow are likely to be different from what had previously been assumed.

TissueMiner can be used in future studies to help understand the molecule signals that influence how cells behave in veins during wing development. The toolkit could also now be used to explore the changes involved in the development of other organs in other organisms.

is required to quantify, interpret and visualize the information derived from segmentation and tracking. Epithelial cells share a set of core behaviors, such as division, rearrangement, shape change and extrusion, which underlie a wide variety of morphogenetic events in different tissues. Methods for analyzing these core behaviors have been developed independently in several labs (*Blanchard et al., 2009*; *Bosveld et al., 2012*; *Etournay et al., 2015*; *Guirao et al., 2015*). However, these analysis tools have not yet been made available to other users in an easy to use and well-documented form.

Here, we propose a generic data layout and a comprehensive and well-documented computational framework called TissueMiner (see *Box 1*) for the analysis of epithelial dynamics in 2D. It enables biologists and physicists to quantify cell state properties and cell dynamics, their spatial patterns and their time evolution in a fast, easy and flexible way. It also facilitates the comparison of quantities within and between tissues. To make TissueMiner accessible to a novice, we provide tutorials that guide the user through its capabilities in detail and release a workflow that automatically performs most of the analysis and visualization tasks we reported previously for *Drosophila* pupal wings (*Etournay et al., 2015*). These tutorials operate using one small example dataset and 3 large wild-type datasets corresponding to the distal wing blade, which we also provide. The code for TissueMiner, along with tutorials and datasets, are publically available (*Box 1*). We illustrate the utility and power of these tools by performing a more extensive analysis of pupal wing morphogenesis focused on differences in the behavior of vein and inter-vein cells.

Wing veins are specified during larval stages, but only become morphologically distinct during prepupal and pupal morphogenesis. During pupal morphogenesis, the dorsal and ventral surfaces of the wing epithelium become apposed to each other on their basal sides, except in the regions that will give rise to veins - here the basal surfaces of dorsal and ventral cells form a lumen. Vein and inter-vein cells also differ on their apical surfaces. Vein cells have a narrower apical cross-section and form corrugations that protrude from the dorsal and ventral surfaces of the wing blade. The cell dynamics underlying vein morphogenesis have never been quantitatively examined.

## Box 1. TissueMiner can be found on the web-based repository GitHub https://github.com/mpicbg-scicomp/tissue_miner#about along with its documentation and tutorials.

Several possibilities are offered to the user to run TissueMiner. For beginners we highly recommend the use of the *docker*, which allows to package an application with its dependencies into a standardized unit for software development (https://www.docker.com/) (*Nickoloff, 2015*). Using a provided docker image for TissueMiner, users can directly run it without any further setup being required. Additional instructions and examples are detailed in the supplementary information and on GitHub. We also provide one example biological dataset that can be used to run TissueMiner tutorials in R. In addition, we give access to 3 databases corresponding to wild-type pupal movies of the distal wing blade. These datasets are available at https://github.com/mpicbg-scicomp/tissue_miner#datasets along with the processed images. Tutorials can be found at https://github.com/mpicbg-scicomp/tissue_miner#documentation.

## Results

We analyze epithelial morphogenesis within TissueMiner in three steps (*Figure 1—figure supplement 1*). First, all epithelial cells of the tissue are digitalized (segmented) and automatically tracked over time using the interactive TissueAnalyzer software (*Aigouy et al., 2010*, *2016*; *Sagner et al., 2012*), which is included in the TissueMiner framework. This software generates segmented images, referred to as segmentation masks that contain information about cell geometry, cell neighbor topology and cell ancestry, which are essential for the study of morphogenesis (*Aigouy et al., 2010*; *Sagner et al., 2012*; *Etournay et al., 2015*). Second, we use a TissueMiner automated workflow to extract this information from the images and store it in a relational database. This workflow also automatically performs most of the visualization steps we describe in this paper (Materials and methods, and Appendix 1). Third, we use TissueMiner's powerful and convenient library of tools for R and Python to query the database to both visualize the data and quantitatively compare cell properties and behaviors between different movies and subregions of the tissue.

Time-lapse datasets are rich with information, and one important set of tools that TissueMiner provides is the ability to visualize this information on the tissue. Such type of visualization can reveal interesting spatial and temporal patterns of core cell behaviors and can guide subsequent analyses. This is, however, insufficient for quantitatively comparing regions within the same tissue or even comparing how the tissue behaves across replicates or various conditions. Therefore, we developed tools to enable the user to define regions of interest, synchronize movies in time, and align all tissues to a common orientation. We then provide tools to easily plot average quantities in different regions or across movies. For each type of measurement, we refer to the tutorials regarding the specific visualization tools we have built (*Box 1*).

### Preparing the dataset (TM R-User Manual sections 1.1 to 1.5)

Before conducting any analysis, the TissueMiner automated workflow reads three configuration files that contain (1) user-defined regions of interest (ROI's), (2) time offsets for movie synchronization, and (3) the rotation angle used to align the tissue to a standard orientation (*Figure 1—figure supplement 1*).

#### Defining regions of interest (howto *Video 1*)

As cellular behaviors may be spatially patterned, one should have the ability to quantify and compare cell dynamics within different ROI's. TissueMiner provides a Fiji macro (draw_n_get_ROIcoord.ijm) to manually define a set of ROI's directly on one given image of the movie. This program manages several lists of ROI's, which the user can create, modify and delete. These lists help maintain the consistency of ROI labels, which is essential for subsequent analysis (*Video 1*).

In addition, defining ROI's of different shapes and following them backwards and forwards in time (*Figure 1A–E'*) is a useful method to visualize tissue deformations (*Figure 1D–D'*,

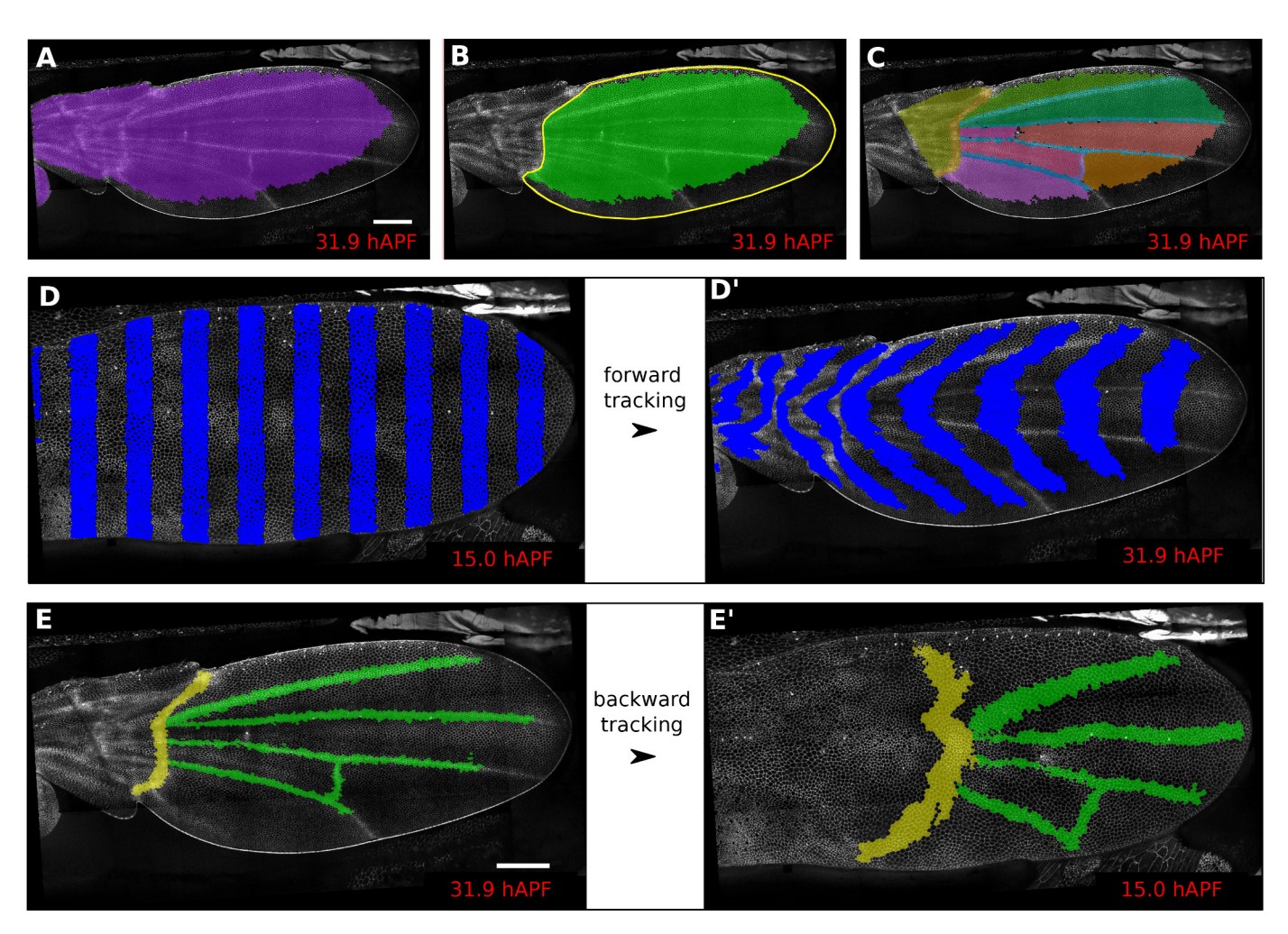

**Figure 1.** Regions of interest are followed in time by browsing the cell lineages. (**A**) Largest population of cells (purple) that remains visible throughout the entire time-lapse. Two cell rows in contact to margin cells were discarded as margin cells are usually not well segmented. (**B**) Largest blade cell population (green) that remains visible throughout the entire time-lapse. The blade region of interest (yellow line) was defined on the last frame of the time-lapse using a custom Fiji macro (https://github.com/mpicbg-scicomp/tissue_miner/blob/master/fiji_macros/). The underlying cell population was then subset using our lineage browser algorithm. (**C**) One can define veins and inter-vein regions of interest and apply the same algorithm as in (**B**). (**D**–**D'**) Regularly spaced regions of interest automatically selected and followed over time to visualize tissue deformation. (**E**–**E'**) Here, we make use of the lineage browser routine to trace back the vein positions at 15 hAPF, as they aren't visible yet at 15 hAPF. Scale bar 50 microns.

The following figure supplement is available for figure 1:

**Figure supplement 1.** Flow chart of TissueMiner.

*Video 2*). These ROI's can be defined at any frame within the movie. Thus, it is even possible to specify a region based on morphological features that only arise late in the morphogentic process under study, which is true of wing veins for example (see *Figure 1E–E'*). ROI definition allows the user to define morphologically relevant regions of interest and compare the behavior of cells in the different regions.

By default, TissueMiner generates two regions of interest – *raw* and *whole_tissue* – in order to select cell populations by name. The *raw* ROI corresponds to all segmented and tracked cells. However cells located at the tissue margin may move in and out of the field of view of the microscope lens. TissueMiner identifies the population of cells (*whole_tissue*) whose entire lineage lies within the field of view throughout the movie. To identify this population, we developed a filtering tool to

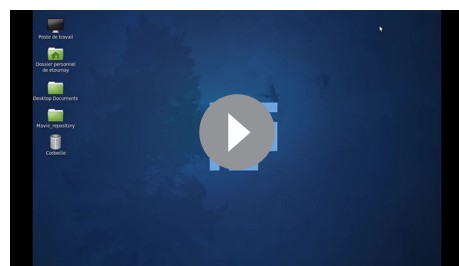

**Video 1.** HOWTO: drawing ROI's.

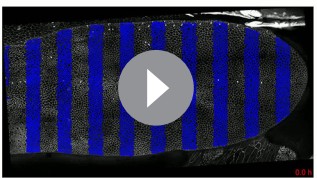

**Video 2.** Visualizing tissue deformation by using vertical stripes.

discard in each movie frame margin cells located at the edge of the segmentation mask and one additional row of cells that contact the margin cells. The choice of discarding two rows of cells is motivated by the fact that segmentation quality drops near the margin. We iterate over all time points to ensure that we discard all cells moving in and out the field of view (see Materials and methods). User-defined ROI's are also subjected to this filtering.

## Aligning movies in time

To temporally align movies, TissueMiner provides a configuration file in which to manually define a time correction for each movie relative to one reference movie whose time correction is set to zero. The time correction can be estimated based on the appearance of morphological landmarks, or by aligning curves of a defined state quantity in time, such as cell area or cell elongation, on the assumption that this quantity has a similar qualitative time evolution.

## Aligning movie orientation (howto *Video 3*)

In order to compare replicates of the same dynamic biological process, all movies should have a common orientation. TissueMiner contains a Fiji macro (orient_tissue.ijm) to assist the user in finding the optimal angle through which each movie should be rotated so that all movies have a comparable orientation (see *Video 3* for an example on the pupal wing).

## Visualizing cell area, cell shape and cell packing on the entire tissue (TM R-User Manual sections 2.2 and 2.6, Py-tutorial sections 2.1 to 2.3)

An important step in analyzing tissue morphogenesis is to quantify cell state properties over time. These properties include cell area, shape anisotropy and packing geometry. In this section, we demonstrate the analysis and visualization tools of TissueMiner by comparing how these state properties evolve during wing morphogenesis in vein and inter-vein regions.

## Cell area and elongation (TM R-User Manual sections 2.2 – 2.5, Py-tutorial sections 2.1 – 2.2)

Morphogenesis is often characterized by changes in cell area and elongation. In the TissueMiner workflow, these properties are calculated from the original segmentation masks and stored in the database (Materials and methods). To visualize the evolution of the cell area pattern at the scale of the whole tissue, we map the area values of each individual cell to a gradient color scale (see *Figure 2A–A'*, *Video 4*). Each cell contour is filled with a color that corresponds to its area. *Figure 2A'* shows the pattern of cell areas in the wing at the end pupal wing blade elongation. This visualization scheme reveals that cells in the proximal hinge and in wing veins have a smaller cross-sectional area (blue) at this time.

Cell elongation is characterized by a nematic tensor describing the axis and magnitude of the elongation (*Aigouy et al., 2010*). As with cell



**Video 3.** HOWTO: Orienting a tissue.

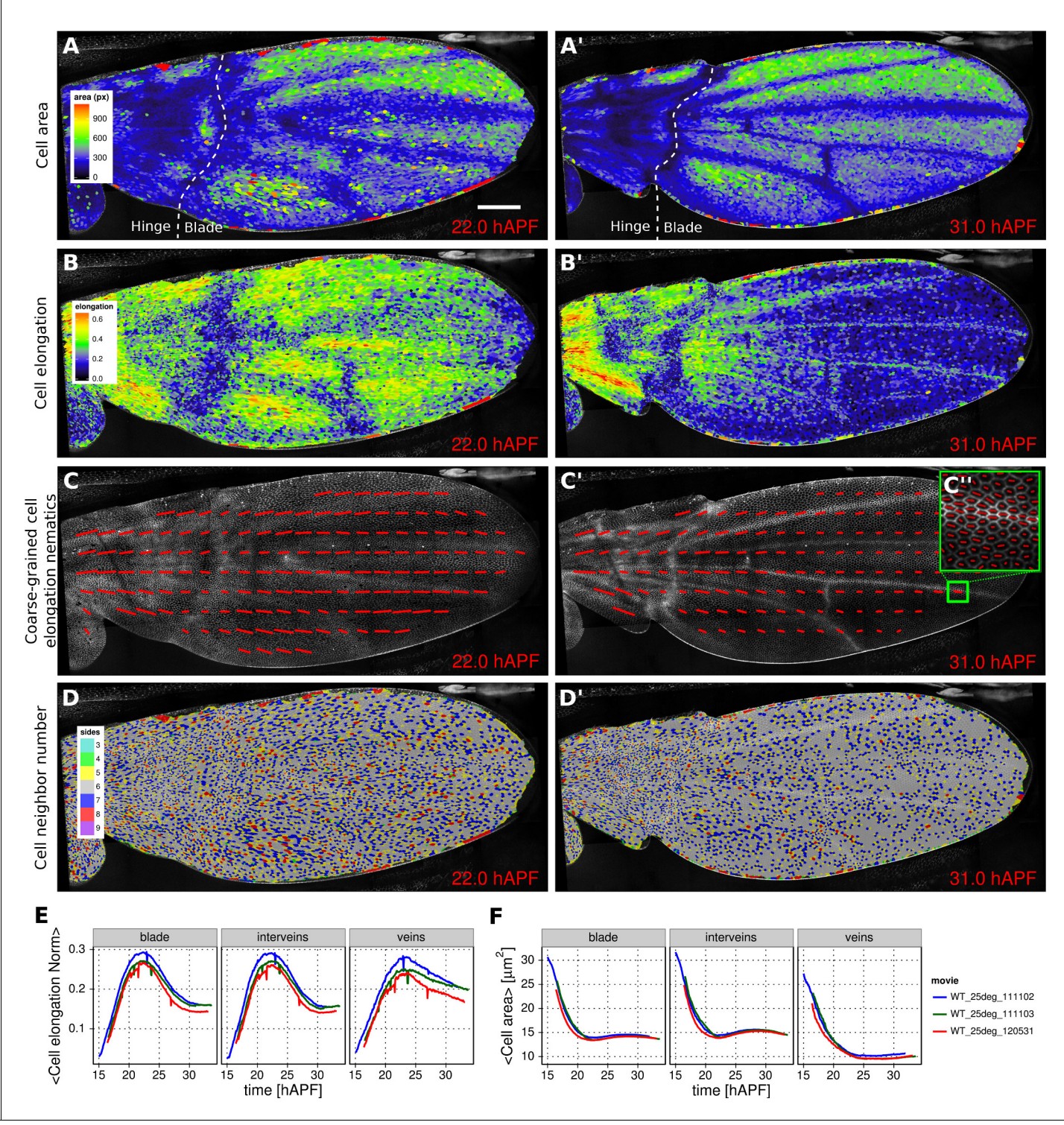

**Figure 2.** Patterned cell state properties in the developing pupal wing of *Drosophila*. (A–D') Cell state patterns at 22 hr and 31 hr after puparium formation (hAPF). (A–A') Color-coded cell area. (B–B') Color-coded cell elongation. The magnitude of cell elongation corresponds to the norm of the cell elongation nematic tensor. (C–C'') Coarse-grained pattern of cell elongation nematics and (C'') cell elongation nematics represented as bars on each individual cell. The wing was divided into adjacent square-grid elements of 33x33 microns in which cell elongation nematics were averaged. (D–D') Color-coded representation of the cell neighbor number. (E) Time evolution of the average cell area in different regions of interest: wing blade (*Figure 1B*), veins (*Figure 1E*), and inter-vein regions. (F) Time evolution of the average cell elongation magnitude in the blade, veins and inter-vein regions. Scale bar: 50 microns.

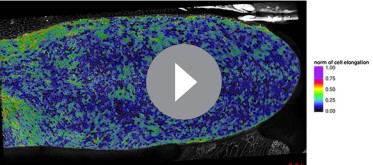

**Video 4.** Color-coded cell area pattern.

**Video 5.** Color-coded cell elongation norm pattern.

area, we map the magnitude of cell elongation to a color scale (*Figure 2B–B'*, *Video 5*). This fine-grained quantification of cell elongation highlights striking differences between inter-vein and vein cells. Inter-vein cells are more elongated than vein cells at 22 hr after puparium formation (hAPF), but this pattern is reversed by 31 hAPF.

The color scale above reveals only the magnitude of the tensor. To visualize both the magnitude and direction of cell elongation, we represent the elongation nematic as a line whose length and angle correspond to the magnitude and angle of cell elongation, respectively. Nematics can also be averaged across multiple cells in a region in order to coarse-grain the patterns and highlight the main features (*Figure 2C–C''*, *Video 6*). For example, the coarse-grained elongation nematics shown in *Figure 2C*, highlight the global alignment of cell elongation in the proximal-distal direction at 22 hAPF.

### Packing geometry (TM R-User Manual section 2.6, Py-tutorial section 2.3)

Cells in the wing become progressively more hexagonal during pupal wing morphogenesis (*Classen et al., 2005*). To visualize packing geometry, we map the neighbor number of each cell to a discrete color code (*Figure 2D–D'*, *Video 7*). This makes changes in packing geometry during morphogenesis immediately obvious (22 and 31 hAPF).

### Plotting temporal evolution of average cell properties (TM R-User manual sections 3.3 to 3.6, Py-tutorial section 3)

The visualization tools described above effectively reveal detailed spatial patterns of cell properties. To highlight how average cell properties change over time, and to facilitate comparison between movies and ROI's, TissueMiner also provides tools to create plots of average quantities as a function of time. In *Figure 2E* and *Figure 2F*, we compare the time evolution of the average cell area and the average cell elongation in movies of the 3 WT wings (blue, green, red) used in (*Etournay et al., 2015*). The plots in *Figure 2* compare the time evolution of average cell elongation and area values for vein and inter-vein cells. We previously showed that average cell area in the wing blade decreases during morphogenesis, but that cell area decrease is balanced by cell divisions to maintain wing blade area. Quantifying average area values in vein and inter-vein ROI's reveals that vein cells contract over a longer period of time than inter-vein cells, and thus have a smaller cross-sectional area at the end of morphogenesis (*Figure 2F*). As previously described, cells in the wing blade elongate and then relax their shapes during pupal wing morphogenesis (*Etournay et al., 2015*) (*Figure 2E*, blade part). Plotting elongation in vein and inter-vein ROI's reveals that vein cells

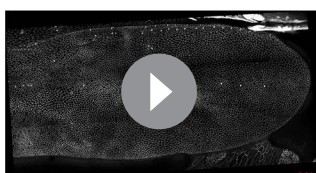

**Video 6.** Coarse-grained cell elongation pattern.

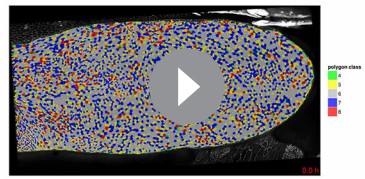

**Video 7.** Color-coded cell packing pattern.

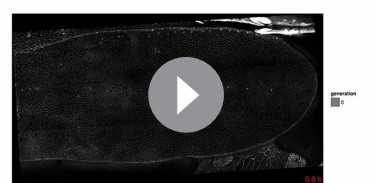

**Video 8.** Color-coded cell generation pattern.

elongate more slowly and also relax their elongation more slowly than inter-vein cells. These differences suggest that vein and inter-vein cells have different mechanical properties.

## Visualizing patterns of cell divisions (TM R-User Manual sections 2.7 – 2.9, Py-tutorial section 2.4)

Oriented tissue morphogenesis may reflect the number, orientation and spatio-temporal pattern of cell divisions. TissueMiner provides several tools to visualize these events. Overlaying color-coded generation number on a pupal wing movie reveals patterns of cell divisions as they occur (*Video 8*), and examining the last frame of the movie (*Figure 3A*) reveals the cumulative pattern of cell divisions. This analysis is largely consistent with the cell division timing inferred from classical BrdU pulse-chase experiments (*Schubiger and Palka, 1987*; *Garcia-Bellido et al., 1994*; *Milan et al., 1996*), but also reveals unexpected additional features. The pattern of cell divisions correlates with veins: most cells in the wing blade divide only once during pupal morphogenesis, whereas in some parts of inter-vein regions they divide twice. These include the cells lying adjacent to veins L3, L4 and L5, and the region posterior to L5. We estimate the median cell-cycle length between the first and second rounds of cell divisions to be (5.25 ± 1.50) hr.

To further investigate how cell divisions are patterned in the blade, we quantified the time evolution of cell division rates in each vein and inter-vein region (*Figure 3B*). This analysis reveals differences in the timing and numbers of cell divisions in these different ROI's. Cells in veins L2 and L4 divide before those in L3 and L5. These divisions are followed by a second peak of division in the inter-vein regions distInterL3-L4, interL2-L3 and postL5 (see cartoon in *Figure 3A*).

To more easily visualize the spatio-temporal pattern of divisions in veins only, the powerful tools available in TissueMiner allow us to assign vein cells a color corresponding to the time at which they divide: blue for 16–18 hAPF and red for 18–20 hAPF (see *Video 9*). This analysis reveals more detailed patterning in division timing. Cell divisions in vein regions that protrude ventrally (L2 and proximal L4), peak at the same time and earlier than those that protrude dorsally (L3, distal L4 and L5). Precise correlation of cell divisions with specific vein and inter-vein regions suggests that they are autonomously controlled by signaling associated with veins.

To measure the orientation of cell divisions, we define a unit nematic tensor (see Materials and methods). For each cell division, the orientation of this unit nematic is defined by the line connecting the centers of mass of the two daughter cells when they first appear (see *Figure 3C–C'*, and TM R-User Manual section 2.8). Each nematic is assigned a position on the tissue that corresponds to the center of combined mass of the two daughter cells. To visualize division orientation patterns, unit nematics can be added within different regions and averaged over different time intervals (*Figure 3D*, *Video 10*, TMR-User Manual section 2.9).

## Visualizing cell junction dynamics (TM R-User Manual sections 2.10 – 2.12, 3.8–3.9)

Epithelial tissues can be reshaped by cell rearrangements, or T1 transitions (for review [*Walck-Shannon and Hardin, 2014*]). In the simplest case, a T1 transition involves two pairs of cells, that exchange neighbors by disassembling one cell-cell contact and replacing it by another – bringing together two previously separated cells (*Figure 4A*). In reality, cell contacts may undergo multiple rounds of shrinkage and regrowth before resolving (*Figure 4A'*). Furthermore some epithelia undergo the related process of rosette formation where multiple cell junctions are disassembled before new neighbors are brought into contact. By separately quantifying the orientation with which cell contacts are gained and lost, one can reveal whether there is a net directionality to cell junction assembly and disassembly. To identify gained and lost cell contacts, we compare cell neighbor relationships between 2 subsequent frames. We exclude changes in neighbor relationships resulting from cell division, extrusion or a cell moving in and out of the field of view. The remaining neighbor relationship changes are used to define cell contacts that have appeared or disappeared.

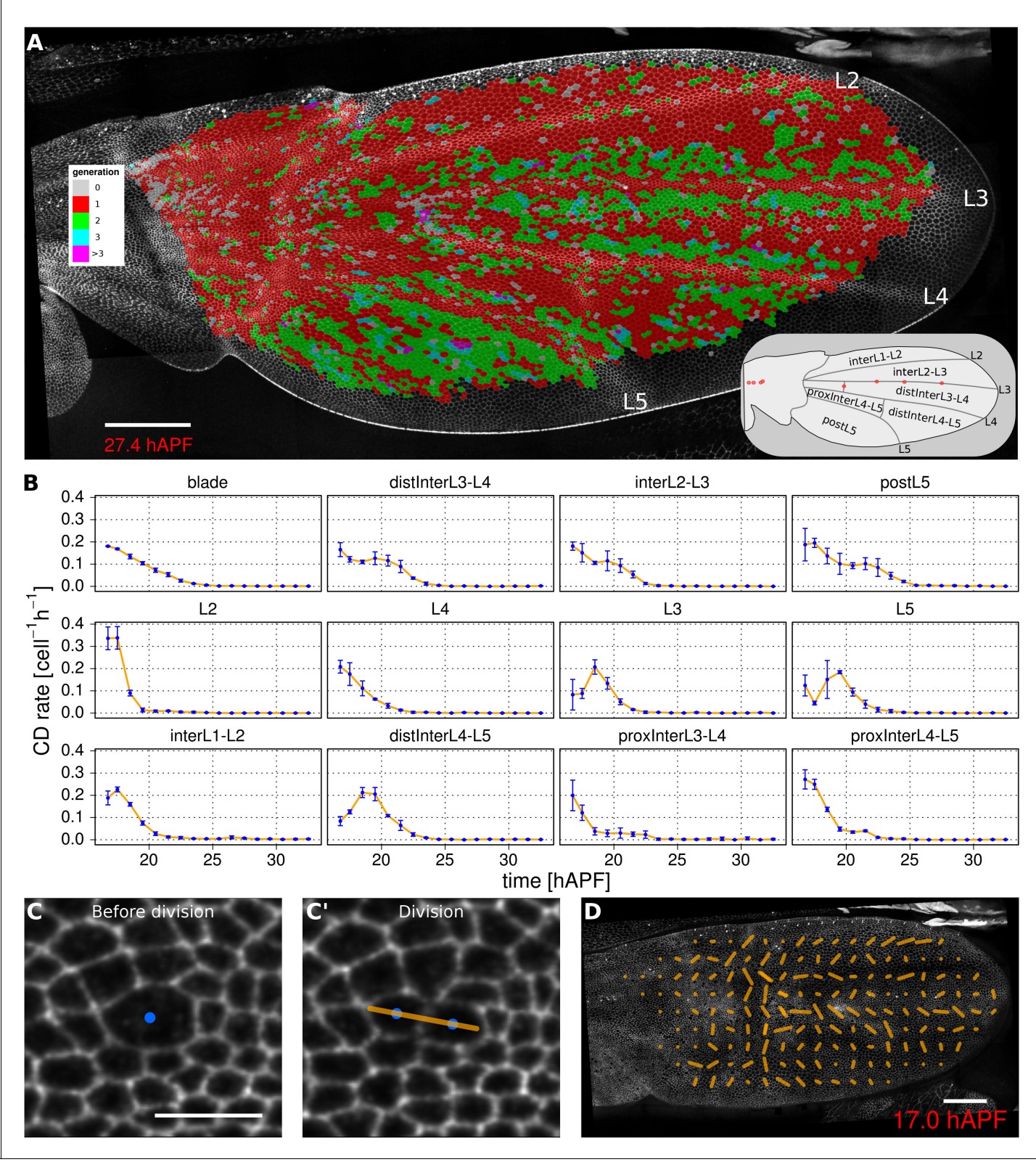

**Figure 3.** Visualization of cell generations and cell divisions. (**A**) Color-coded pattern of cell generations. The wing cartoon on the bottom right shows the names of subregions that we analyze in panel B. Scale bar 50 microns. (**B**) Cell division rate in different regions of interest. To smooth fluctuations, these rates were averaged in discrete time intervals of one hour (TM R-User Manual, section 3.7). We further averaged these rates amongst the three wild-type wings. Error bars depict the standard deviation between wings. Cells divide earlier in veins L2 and L4 than in L3 and L5. Two maxima

*Figure 3 continued on next page*

*Figure 3 continued*

corresponding to two rounds of divisions are visible in inter-vein regions: interL2-L3, distInterL3-L4 and postL5. (**C–C'**) A dividing cell with its unit nematic depicting the division orientation. Scale bar 10 microns. (**D**) Coarse-grained pattern of cell division orientation (grid size of 33x33 microns). Scale bar 50 microns.

We characterize the orientation of contact gains and losses by assigning them a unit nematic tensor. For contact loss, the orientation of the nematic is defined by the axis intersecting the two cell centers. For contact gain, the orientation of the nematic is perpendicular to the axis intersecting the two cell centers (*Figure 4A–A'*). If there is a simple disappearance and reappearance of a single cell contact, corresponding nematics will cancel out. Therefore, the sum of contact gain and contact loss nematics over time and/or space will represent an effective T1 nematic describing net direction of contact assembly/disassembly.

The rate of contact gain and loss can be visualized in different ways. Cell contact dynamics can be viewed directly on movies of tissue morphogenesis by assigning colors to cells as they gain (red) or lose (green) contacts. Those cells that simultaneously gain and lose different cell contacts are colored blue (*Figure 4B–B'*).

The frequency of contact gain and loss, independent of orientation, can be plotted over time. *Figure 4C* compares the frequency of contact assembly/disassembly in vein and inter-vein regions. In both regions, this rate begins to decrease in the second half of morphogenesis.

To visualize the pattern of orientation of T1 transitions throughout the wing, we sum contact gain and loss nematics over square grid elements, and average over a chosen time window (about 50 min in *Figure 4D*, *Video 11*, see TM R-User Manual section 2.12).

Finally, the average orientation of effective T1 nematics in sub-regions over time can be visualized using circular diagrams, where nematics are color-coded to indicate developmental time. *Figure 4—figure supplement 1A* reveals that the orientation of effective T1's is along the anterior-posterior (AP) axis early (blue) and shifts to the proximal-distal (PD) axis in the second half of morphogenesis (red). A similar approach can be used to illustrate average cell elongation nematics over time (*Figure 4—figure supplement 1B*).

## Quantification of tissue deformation and the contribution of different cellular events (TM R-user manual section 3.10)

While it is useful to quantify the number and orientation of cellular events like elongation, rearrangement, extrusion and division, this by itself does not provide quantitative information about the amount of tissue shape change contributed by each type of event. We therefore devised a method to measure deformation caused by these cellular processes such that they sum to the measured tissue deformation.

Tissue deformation can be decomposed into isotropic and anisotropic parts that distinguish changes in area (compression/expansion) from changes in aspect ratio (pure shear, for details see also Materials and methods). The quantities describing area changes are scalar, whereas the quantities describing shear rate in a 2D-network are nematic tensors harboring two distinct components that describe the orientation and magnitude of the shear.

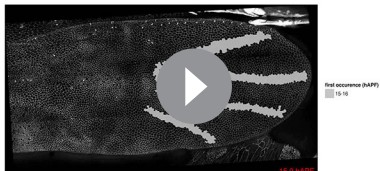

**Video 9.** Color-coded cell division pattern in veins and by time intervals.

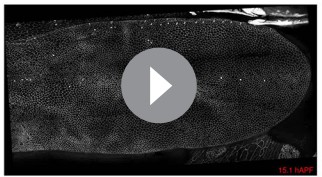

**Video 10.** Coarse-grained cell division pattern.

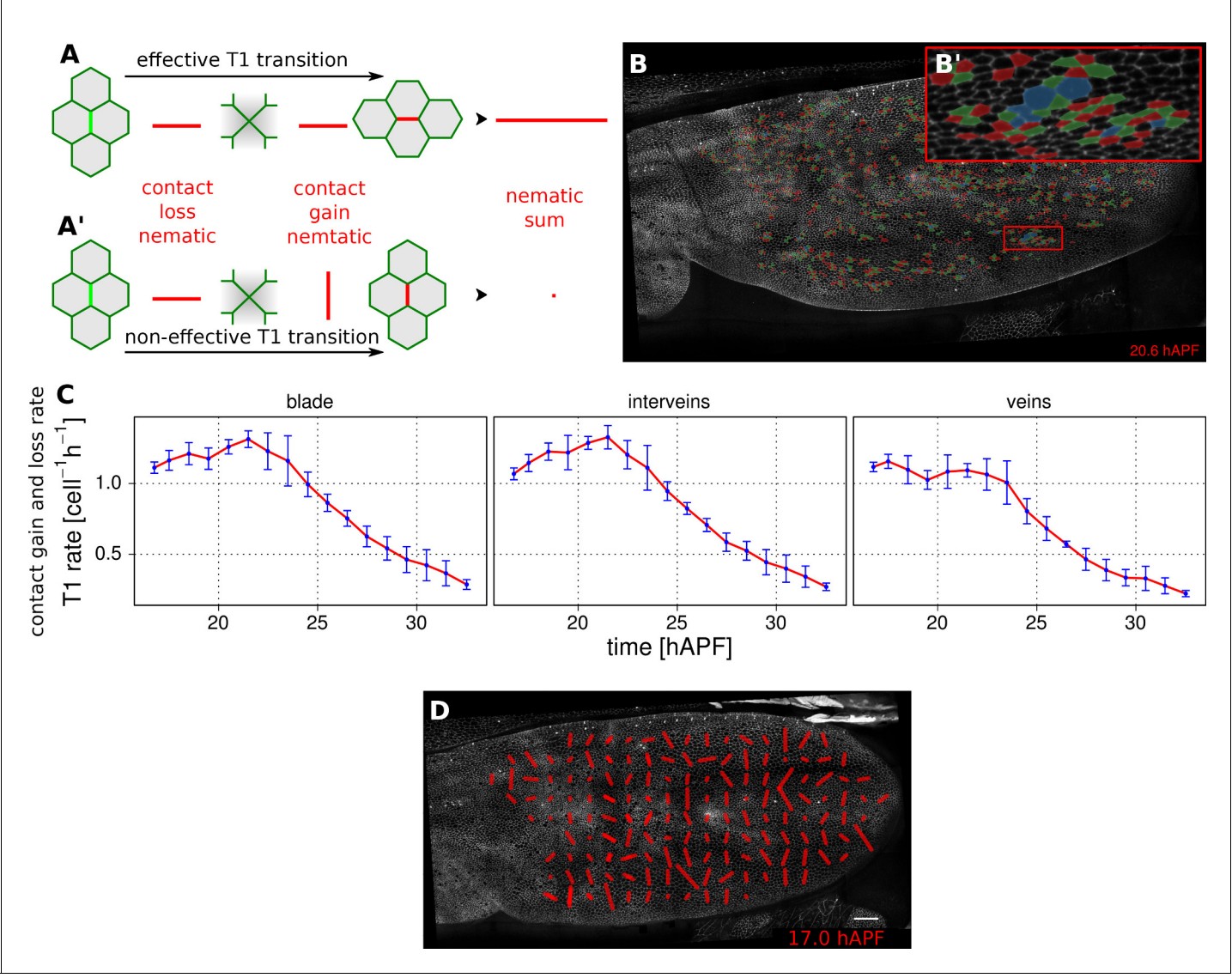

**Figure 4.** Visualization and quantification of T1 transitions. (**A–A'**) Cartoon depicting an effective T1 transition (**A**) that corresponds to cell-contact loss and gain in different directions. Each contact loss or gain is assigned a unit nematic describing its orientation. (**B–B'**) Pattern of cells losing contact (green), gaining contact (red) or both (blue). (**C**) Rate of neighbor change per cell and per hour in the blade, veins and inter-vein regions of interests. Rates were averaged within discrete time intervals of one hour and further averaged among the 3 WT wings (TM R-User Manual, section 3.8). Error bars depict the standard deviation amongst wings. (**D**) Coarse-grained pattern of neighbor exchange orientation at 17 hAPF. Cell neighbor change nematics were obtained by summing up unit nematics in each grid elements of 33x33 microns and further averaged in time using a 50 min time window. Scale bar 50 microns.

The following figure supplement is available for figure 4:

**Figure supplement 1.** T1 and cell elongation nematic orientation.

Tissue area changes can be calculated based on cell area change and the number of cells gained and lost by divisions and extrusions – information that is all available in the TissueMiner database (*Etournay et al., 2015*).

To quantify the cellular contributions to anisotropic tissue deformation, TissueMiner uses the so-called Triangle Method, which is based on a triangular tiling of the junctional network (*Etournay et al., 2015*; *Merkel et al., 2016*). Triangle elongation is a proxy for cell elongation, and topological changes in the network result in redrawing of triangles (*Figure 5A–C*). The resulting

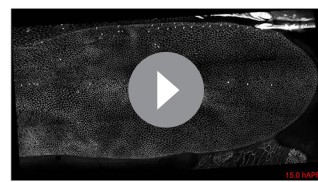

**Video 11.** Coarse-grained cell rearrangement pattern

change in average triangle elongation can be used to calculate the shear due to the topological changes (*Etournay et al., 2015*). In addition to contributions from divisions, cell rearrangements, extrusions and cell shape changes, the method also takes into account deformation caused by correlations between elongation and both area change and rotation.

## Validation of tissue deformation measurements using computer-generated cells

To test the reliability of TissueMiner in calculating large cell and tissue deformations, we created two computer-generated movies of hexagonal cells sheets (*Videos 12*, *13*). In one movie, we imposed a constant isotropic expansion rate of $3.50 \times 10^{-2}$ per frame, without any anisotropic deformation. In the second movie, we imposed a constant pure shear along the x-axis with a rate of $1.75 \times 10^{-2}$ per frame, and without any isotropic expansion. The amounts of isotropic expansion and pure shear have been chosen to be at least 10 times higher than what we measure between subsequent frames of pupal wing movies.

We then asked if TissueMiner could quantitatively recapitulate the respectively imposed deformation rates. In each dataset, TissueMiner automatically defines a 'whole_tissue' region of interest that corresponds to a consistent set of cells that are always visible (about 100 cells in the isotropic expansion movie and about 50 cells in the pure shear movie, green labels in *Videos 12* and *13*). All measurements are done in this ROI to avoid measuring deformation due to inward and outward cell flows. *Figure 5—figure supplement 1* shows the time evolution of the measured tissue expansion rate (panel A) and tissue shear rate (panel C) that were averaged over the 'whole_tissue' ROI, and their respective cellular contributions. Panels B and D show the corresponding cumulated curves. As expected, in the isotropic expansion movie we observe a nearly constant isotropic expansion rate, which is accounted for by the cell area change contribution. We measure an average expansion rate of $(3.53 \pm 0.04) \times 10^{-2}$ per frame, which is consistent with the value imposed when creating the movie. The measured uncertainty is the 95% confidence interval of the standard error of the mean. The pure shear rate and its cellular contributions nearly vanish in this movie (*Figure 5—figure supplement 1C,D*).

For the pure shear movie, we measure an approximately constant horizontal component of the pure shear rate of $(1.74 \pm 0.02) \times 10^{-2}$ per frame, which is consistent with the value imposed when creating the movie. This pure shear rate is entirely accounted for by cell elongation change. The isotropic expansion rate and its cellular contributions nearly vanish (*Figure 5—figure supplement 1A, B*). Other contributions to expansion and shear rates are negligible in both movies.

The pixelated nature of individual cell contours contributes to fluctuations of our measured values. Moreover, we find that these fluctuations cancel out when cumulating the deformation (*Figure 5—figure supplement 1B and D*). Thus, the current implementation of TissueMiner captures the tissue isotropic expansion and pure shear rates as well as the corresponding cellular contributions with a good precision in these computer-generated movies.

## Deformation of the pupal fly wing

*Figure 5—figure supplement 2* shows the rate of relative area change and cumulative area change of vein and inter-vein regions over time, as well as the cellular contributions to these area changes. As previously noted, the area of the blade as a whole changes very little. However sub-region analysis reveals that inter-vein expansion compensates for compression in vein regions. Vein cells not only divide less than inter-vein cells, but also decrease their area more.

Next we use the Triangle Method to calculate pure shear rates in the time-lapse movies of developing pupal wings. To visualize the spatial pattern of pure shear rate in the wing, TissueMiner allows us to plot nematics corresponding to the local tissue shear rates (*Figure 5D*) and to rates of shear produced by different cellular contributions (*Figure 5—figure supplement 3*, and [*Etournay et al., 2015*]) averaged within the squares of about 26 x 26 microns.

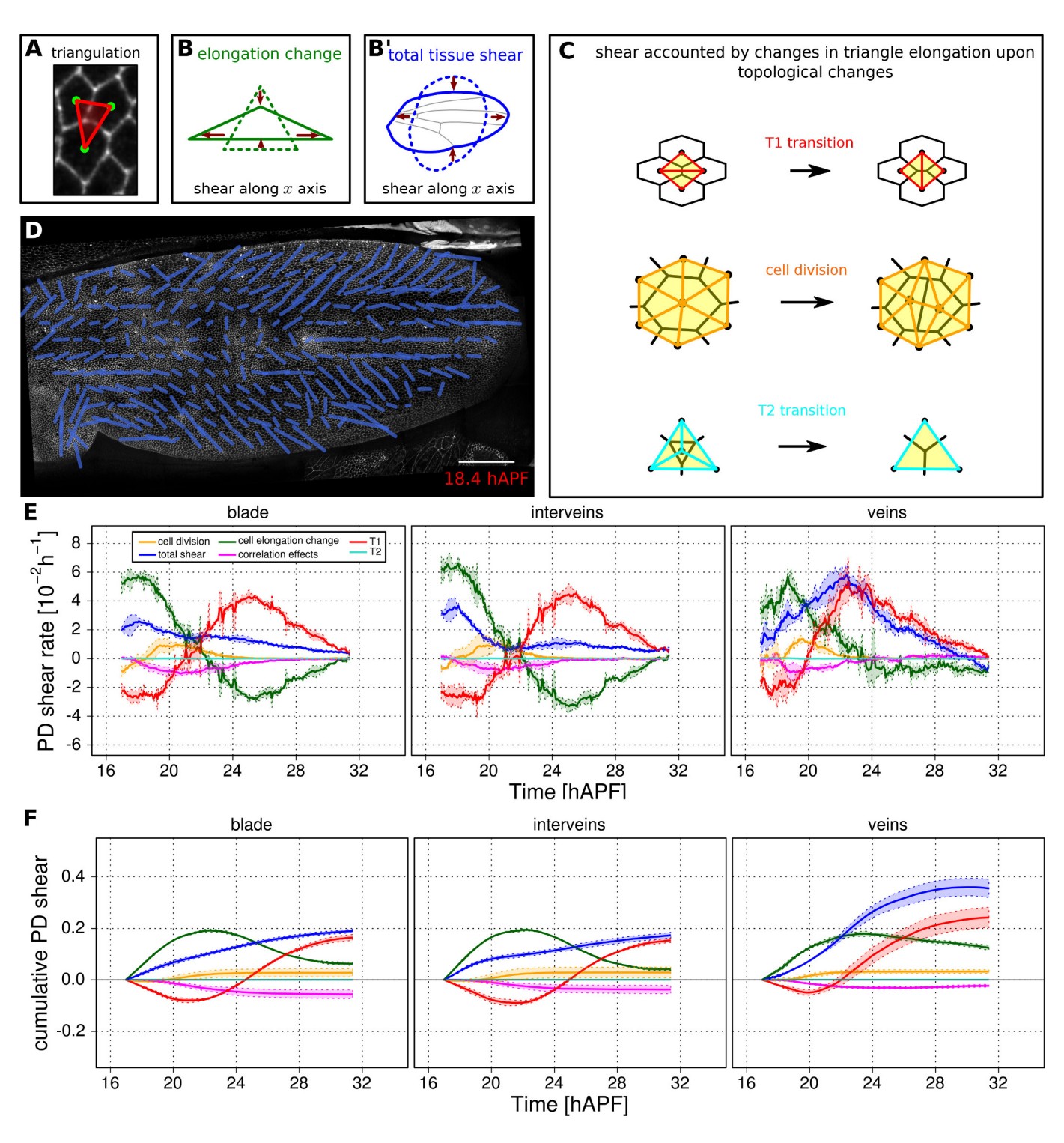

**Figure 5.** Visualization and quantification of anisotropic cell and tissue deformation. (A) Triangulation of the cell network: each triangle vertex corresponds to a cell center. (B–B') Cartons depicting triangle pure shear and total tissue shear along the x axis. (C) Cartons depicting shear due to T1 transition, cell division and extrusion. (D) Pattern of local tissue shear rate obtained from the triangulation method. Scale bar 50 microns. (E) shows the average rate of tissue shear (blue) in the blade, interveins and veins, and the corresponding cellular shear contributions (other colors). Shaded regions indicate the standard deviation amongst wings. (F) shows the accumulated tissue shear over time and the accumulated contributions of each type of cellular event. The tissue shear (blue) in veins is orientated along the PD axis and it is higher than in inter-vein regions during most of pupal morphogenesis. It leads to an extension along the PD axis and to a narrowing along the anterior-posterior (AP) direction. By the end of the movie,

*Figure 5 continued on next page*

*Figure 5 continued*

accumulated tissue shear (blue) is almost twice as high in veins as in inter-vein regions. Shaded regions represent the standard deviation amongst wings.

The following figure supplements are available for figure 5:

**Figure supplement 1.** Measurements of cell and tissue deformation from two computer-generated sheets of hexagonal cells.

**Figure supplement 2.** Tissue isotropic deformation and cellular contributions in different regions.

**Figure supplement 3.** Comparison of patterns of cell event orientation with their correponding quantitative patterns of shear.

To compare the time evolution of pure shear rate between different tissue subregions we plot this rate averaged over the corresponding ROI (*Figure 5E–F* and [*Etournay et al., 2015*]). A positive sign for shear indicates an extension along the PD axis and a contraction along the AP axis, whereas a negative sign indicates an extension along the AP axis and a contraction along the PD axis.

As reported previously, the wing blade as a whole shears along its PD axis between 16 and 32 hAPF. T1 transitions and cell elongation are major contributors to total PD shear, and they display complementary behavior that evolves over time. In the first phase, cells elongate in the PD axis in response to tissue stresses generated by hinge contraction, and by actively oriented T1 transitions that occur first along the AP axis. In the second phase, cell elongation causes the orientation of T1 transitions to shift 90° from the AP to the PD axis (*Etournay et al., 2015*). These PD oriented T1 transitions both contribute to tissue shear and relax PD cell elongation. We now compare shear and cellular contributions to shear in vein and inter-vein regions. Tissue shear peaks earlier in inter-vein regions than in veins, but veins shear more overall. Examining the cellular contributions to shear suggests that increased shear in veins reflects a different relationship between cell elongation and T1 transitions. PD-oriented T1 transitions do not only produce more shear in veins, they also fail to relax PD cell elongation as much as in inter-vein regions.

## Discussion

Quantitative image analysis of developing epithelia is a powerful approach to understanding morphogenesis, but the tools with which to tame and analyze these complex data have not been widely available in a standard and well-documented format. Here we provide an introduction to the capabilities of TissueMiner and tutorials for its use. TissueMiner provides general strategy to store and analyze large data sets of interwoven objects by combining state of the art tools for data mining. It allows quantification and visualization of epithelial morphogenesis at multiple scales – from individual cells to entire tissues. It provides both a generic database format and a multi-platform toolkit to interrogate and visualize data and quantify cellular contributions to large-scale epithelial deformations.

TissueMiner has been designed to be versatile and expandable. The database format we provide standardizes the organization of tracked cell data and collects all data into a single file per movie. Such a standardized data format facilitates data sharing between different sources, thereby enhancing cross-laboratory reproducibility. As the database stores positional information about cells and

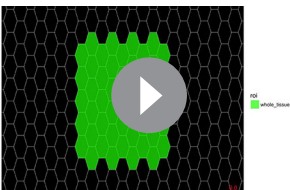

**Video 12.** Computer-generated hexagonal cells with an imposed shear rate.

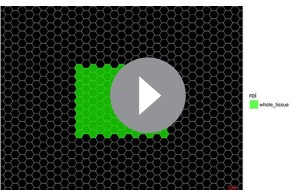

**Video 13.** Computer-generated hexagonal cells with an imposed isotropic expansion rate.

cell contacts, as well as cell neighbor topology, it could also be useful for parameterizing simulations of epithelial remodeling by vertex models or other physical network models. The scheme of our relational database is expandable: additional properties of cells, bonds and vertices can be appended to the database without affecting the relationships between tables. As a consequence, our current query tools to interrogate the database remain functional, even if the database is extended with new properties of cells, bonds and vertices.

TissueMiner takes advantage of the advanced graphical capabilities of R and Python to enable the visualization of patterns of deformation and cell state properties directly on the movie images or quantitatively summarized in graphs. In particular, R provides a flexible grammar with which to manipulate tables obtained from the database and to easily plot graphs (*Wickham, 2009*; *Francois and Wickham, 2015*). TissueMiner also offers multiple options for coarse-graining data in space and time through an expandable collection of scripts, which constitutes the TissueMiner library for R or Python. These two easy-to-learn programming languages give TissueMiner its great flexibility to both address general questions of epithelial morphogenesis and project-specific questions, and enable automation, parallelization and customization of user-specific workflows.

The tools underlying TissueMiner were originally developed to understand the interplay of cell dynamics and epithelial tension on the developing wing of the fruit fly, where we described cellular contributions to pupal wing morphogenesis averaged throughout the entire wing blade (*Etournay et al., 2015*). Here, to illustrate the utility of the TissueMiner framework, we compared the behavior of vein and inter-vein regions in the developing pupal wing. Comparing cell dynamics in veins and inter-vein regions provided an unexpected explanation for the process of 'vein refinement'. Vein refinement refers to the fact that veins become narrower during pupal morphogenesis. This had been interpreted as a signaling-dependent reduction in the number of cells assuming the vein fate (*Blair, 2007*). Here we show instead that vein narrowing results from a convergent extension-like process that is stronger in veins than in inter-vein regions. This elongates and narrows the veins without reducing vein cell number. It will be interesting to examine how signaling pathways involved in vein refinement influence cell dynamics in veins during morphogenesis. The standardization of analysis that TissueMiner provides will facilitate these and other comparisons critical for deciphering the molecular mechanisms underlying epithelial morphogenesis.

## Materials and methods

### Live imaging of the pupal wing

The knock-in Ecad::GFP fly line (*Huang et al., 2005*) was used for live imaging of the developing pupal wing. Flies were raised and maintained at 25°C during imaging by using a temperature-controlled chamber equipped with a humidifier to prevent desiccation. Long-term time-lapse imaging was performed as previously described (*Etournay et al., 2015*). After the imaging session, flies were maintained in a humid environment where they eclosed at the term of pupal development.

### A relational database to store the history of cells, their lineage and their constituent bonds and vertices

The visualization and quantification of cell dynamics underlying tissue morphogenesis rely on the ability to extract information about cell geometry, cell neighbor topology and cell histories from time-lapse movies (*Aigouy et al., 2010*; *Etournay et al., 2015*). We use TissueAnalyzer to segment and track the cell network over time. This results in a series of digital images that contain this information (*Figure 6—figure supplement 1*). To facilitate its access and use, we developed tools in the TissueMiner framework to extract and convert this information initially stored in images into a specific database format (see details in appendix 1), which we call 'TM-DB' (schematically outlined in *Figure 6A*).

First, the history of each tracked cell in the movie is stored as a separate row in the *cell_histories* table of the TM-DB (*Figure 6A*). This includes the movie frames in which it first appears and disappears and why, along with its lineage relationship to other cells (see appendix 1). The reason for cell appearance and disappearance is inferred by the parser. A primary reason could be a cell division, which results in the disappearance of the mother cell and in the appearance of two daughter cells. It could be a cell extrusion that results in its disappearance. It could also be that cells move in and out

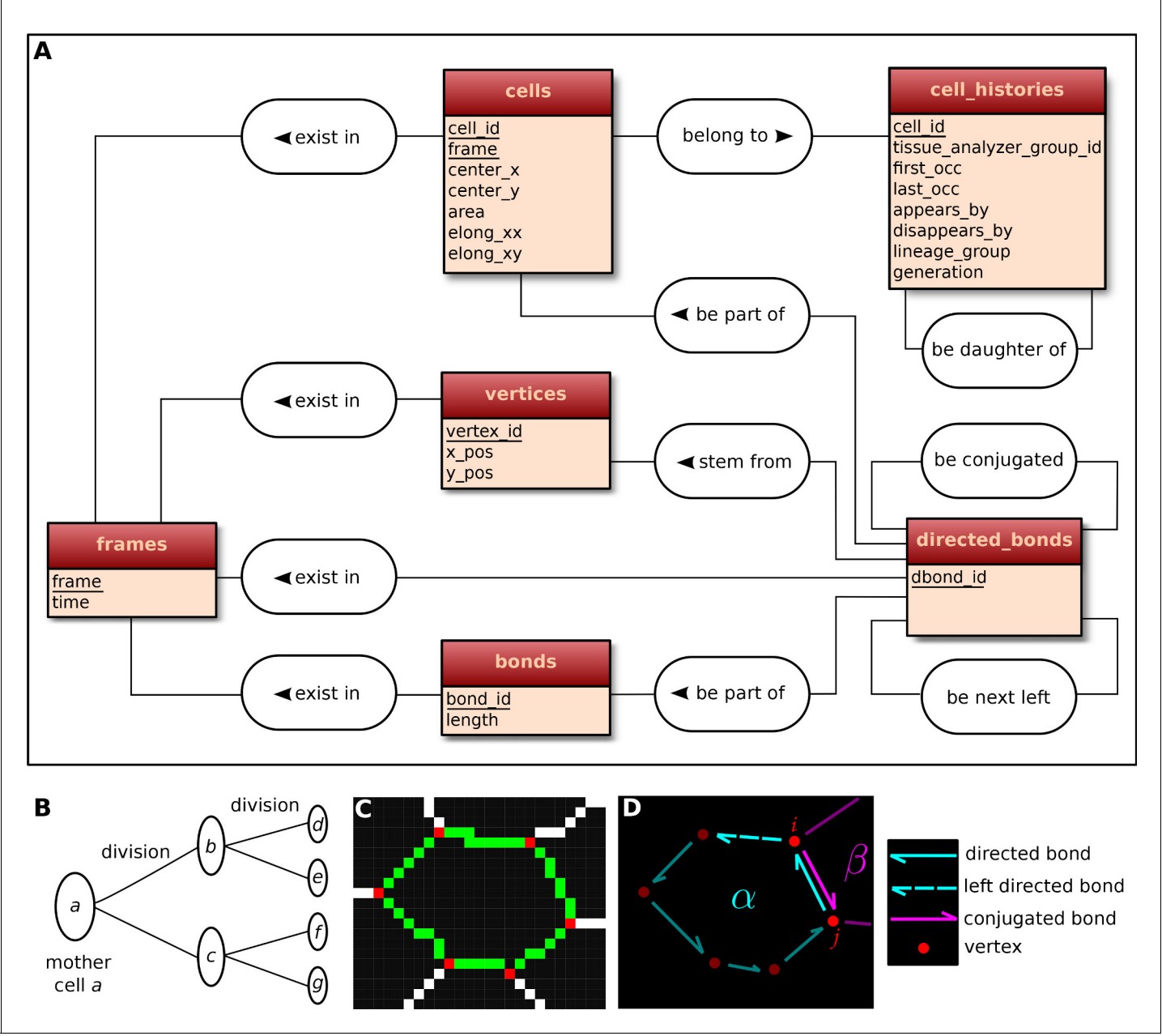

**Figure 6.** Construction of the relational database of TissueMiner. (**A**) Conceptual scheme of the database. Entities (square boxes) are related to other entities by associations (rounded boxes). Each entity contains an identifier (underlined) that uniquely defines each record. The database can be implemented by converting entities into tables (see appendix 1 and *Figure 6—figure supplement 2*). (**B**) Cell lineage trees are stored in the database: upon division a mother cell identifier a gives rise to two new daughter cell identifiers b and c. {a,b,c,d,e,f,g} defines one lineage group. (**C**) A pixelated cell contour in the 2D cell network: green=bond pixels, red=vertex pixels, white=other cell network pixels. (**D**) Vectorized representation of the cell shown in (**C**). To preserve the topology of the cell network, directed bonds (cyan) are defined from within a given cell alpha and ordered anticlockwisely along the cell contour. Each directed bond is complemented by a conjugated bond (magenta) and is linked to it next counter-clockwise follower (dashed).

The following figure supplements are available for figure 6:

**Figure supplement 1.** Tracked cells identified by unique colors in TissueAnalyzer.

**Figure supplement 2.** Logical scheme of the relational database.

of the field of view of the microscope lens, resulting in gain and loss of cells. Furthermore, we use this information to establish the lineage relationship that corresponds to each group of cells related by ancestry (*Figure 6B*). Each cell within the lineage group is assigned a generation number. The lineage group and generation number for each cell are listed in the *cell_histories* table.

We store the time points at which the movie images were recorded into a *frames* table that links each movie frame to its corresponding time point. For each movie frame, we need to store geometrical and topological information about cells within the cellular network. Geometrical information includes position and shape descriptors, whereas topological information indicates the arrangement of neighboring cells around each cell. We use cell histories, geometry and topology to understand how individual cells contribute to the whole tissue deformation during morphogenesis (*Etournay et al., 2015*).

The geometrical information is stored in three tables of the TM-DB: *cells, bonds* and *vertices.* They correspond to the 3 generic entities - cells, cell-cell contacts and intersections between cell-cell contacts, respectively illustrated in *Figure 6C*. These entities are commonly used in vertex model simulations (for review [*Fletcher et al., 2014*]). The *cells* table contains cell geometrical data (center of mass, area, shape anisotropy) and the polarized distribution of proteins along the cell circumference, as represented by a polarity nematic tensor (*Aigouy et al., 2010*). The *bonds* table informs about bond length, and the *vertices* table about vertex position in each movie frame.

The *directed_bonds* table exclusively stores the cell neighbor topological information at each frame, *i.e.* how bonds are organized around each cell along with the cell neighbor relationship information. To store the cell neighbor topology in an unambiguous manner, we define for each cell a directed path of consecutive bond vectors oriented counterclockwise, which forms the oriented circumference of the cell (*Figure 6D*, see also [*Kachalo et al., 2015*]). We link each directed bond to its counter clockwise follower (*left directed bond*) in the same cell. To store the cell neighbor relationship, we link each directed bond to its corresponding directed bond (*conjugated bond*) of the neighboring cell (*Figure 6D*, and appendix 1).

The TM-DB is relational, which means that it establishes contextual relationships between items stored in one ore more tables (see appendix 1). These relationships are outlined in rounded boxes in the conceptual scheme of the TM-DB (*Figure 6A*). Technically, each item in a table is stored in a separate row and is given a unique number as identifier. For a relationship between two tables, one of the tables contains an additional column, which refers to items in the other table by holding their identifier number. Such additional columns for the TM-DB format are shown in blue in *Figure 6—figure supplement 2*. When extracting information from a database using so-called queries, these columns serve as bridges connecting the information stored about related items.

In essence, this structure creates a generic relational model to represent complex cell tracking data in 2D. In practice, the data for each movie is stored in a separate SQLite database file. Since all movie files are stored using the same database structure, automated data mining and visualization are greatly facilitated. For the same reason, usage of the TissueMiner database format encourages exchange of both movie data and analysis tools.

## An automated workflow compliant with high performance computing platforms

To help the user to perform complex tissue morphogenesis analysis, we developed an automated pipeline that uses the tracked data from TissueAnalyzer as an input to build the database and perform all downstream analyses described above. To do so, we use the *snakemake* workflow engine developed by Koster and Rahmann (*Koster and Rahmann, 2012*). This engine channels the different processing steps into a well-formed workflow graph. *Snakemake* automatically determines the execution order, provides means for error recovery and job control, and supports High Performance Computing (HPC) environments. By using *snakemake* we enable the user to easily run and monitor TissueMiner, while maintaining a proper decoupling of tools as independent executables.

Practically, the user defines a workflow definition file in which processing steps are defined as a set of execution rules, namely a list of scripts to be run along with required input(s) and expected output(s). *Snakemake* automatically builds a directed graph from which the execution order of processing steps is inferred. If only one branch of the graph needs to be run, the engine will ensure that all input data are present and will automatically run upstream steps if necessary. This engine also

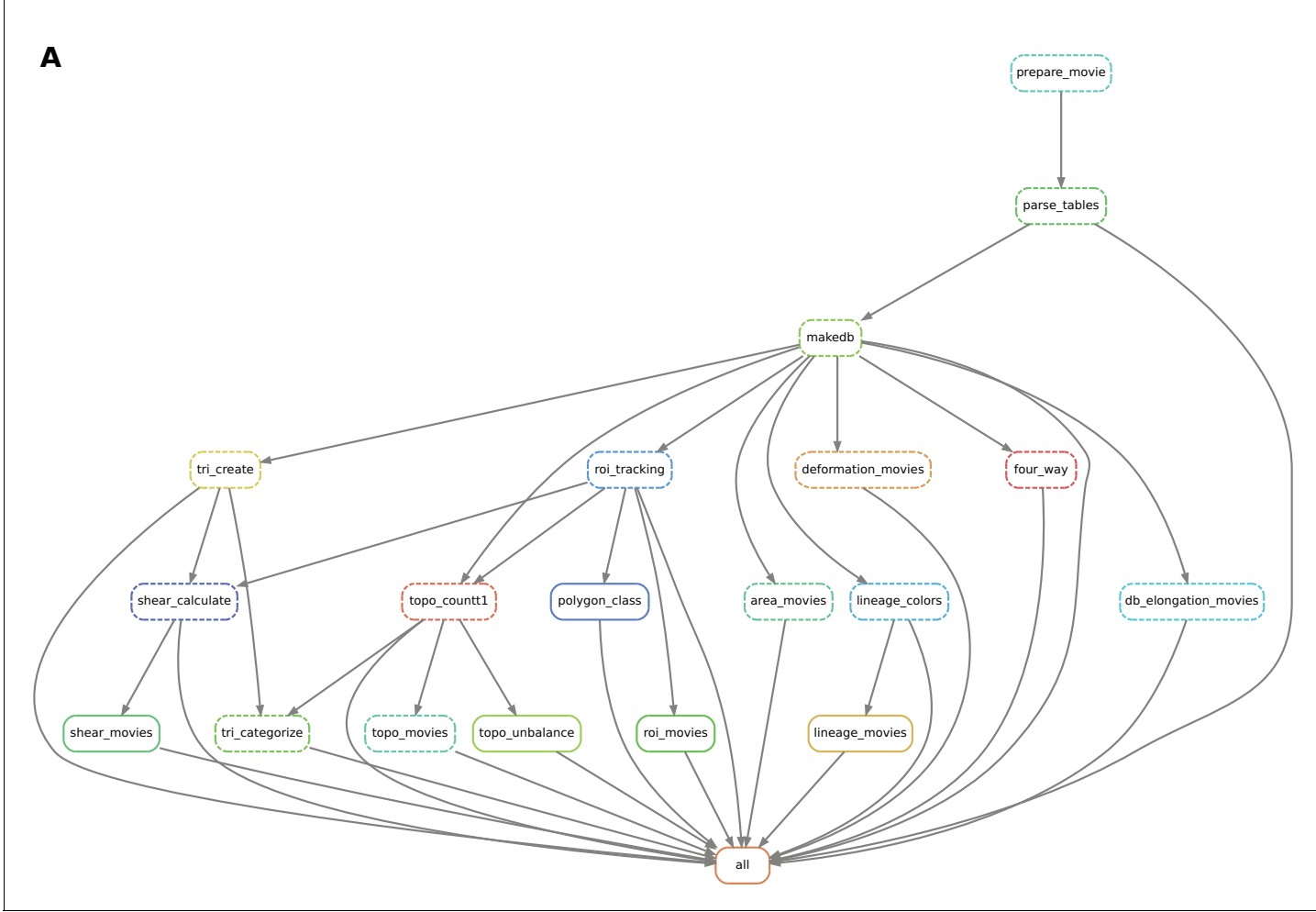

**Figure 7.** Automated workflow using snakemake. (**A**) The snakemake engine can generate a directed acyclic graph (DAG) where we show an example here. This graph represents both the execution dependency (grey arrows) and the execution state of the workflow (solid or dashed line). Each box corresponds to an execution rule, namely a program to be run along with required input(s) and expected output(s). This DAG can be generated at any time when running the workflow (see documentation). Solid lines indicate the rules that have not been executed yet, whereas dashed lines depict completed jobs. The first rule to be executed is called 'prepare_movie': it prepares the tracked images from TissueAnalyzer to be converted by the parser into tables of values containing all necessary entities along with their properties ('parse_tables' rule). Then the 'make_db' rule is executed for building the database. Following the grey arrows can one navigate into the next steps of the workflow. The 'roi_tracking' rule filters out cells in contact to margin cells including user-defined regions of interest, and the 'roi_movie' rule allows us to visualize regions of interest over time. The 'deformation_movies' and 'db_elongation_movies' rules generate annotated movies to visualize the deformation of the tissue and the cell state properties (area, elongation). The 'four_way' rule detects four-way vertices and performs basic statistics on vertices. The 'tri_create' rule performs the triangulation of the network for further shear calculation and visualization ('shear_calculate' and 'shear_movies'). It also enables triangle tracking and mapping to each type of cell event ('tri_categorize'). The 'topo_countT1' rule detects neighbor changes that are not due to division or extrusion, and categorizes them into *gained* or *lost* neighbors. The 'topo_movies' rule allows one to visualize the coarse-grained rates of division and neighbor changes on the tissue. The 'topo_unbalance' rule is a quality check to verify that the number of gained neighbors is similar to the number of lost neighbors. The 'polygon_class' rule performs the cell-neighbor number count. The 'lineage_colors' rule allows us to optimize the color of each lineage group such that adjacent lineage groups always have different colors. Finally, the 'lineage_movies' rule allows one to visualize lineage groups and cell generations on the tissue. The rule 'all' checks that all upstream jobs have been completed.

provides the possibility to visualize a directed acyclic execution dependency and execution state graph (DAG) for a given workflow (see *Figure 7*).

One major advantage of a workflow engine such as *snakemake* is that it can run the workflow on various architectures - from single-core workstations to multi-core servers and clusters - without the need to modify the rules, thereby facilitating reproducible research. To simplify the TissueMiner

installation procedure, we provide a pre-configured system to be loaded in the *docker* software available at http://docker.com. The TissueMiner docker image can be run without any setup using provided example data or custom user data as detailed out on the TissueMiner *GitHub* project page. More advanced users can use TissueMiner directly from the command-line with or without *snakemake* and can thus perform simultaneous analyses of multiple movies.

## A user-friendly data-mining library to easily collect information for comparing multiple datasets

After applying our automated workflow to different movies, the results can be easily compared using a collection of command-line tools written in R and Python. These tools aggregate different experiments for plotting and performing comparative analysis. Here we describe the tools written in R. Python tools are described in the corresponding tutorial. The R tools are designed to be used in an integrated development environment such as RStudio, which provides a user-friendly environment to assist the user in writing and executing command lines. These command line tools are organized in the spirit of a grammar of data manipulation and they can be combined with the existing R tools like dplyr (*Francois and Wickham, 2015*) or ggplot2 (*Wickham, 2009*) for manipulating and visualizing data (https://mpicbg-scicomp.github.io/tissue_miner/user_manual/Learning_the_R_basics_for_TissueMiner.html).

We developed generic 'multi-query functions' (*mqf*) to collect specific information for individual movies. These *mqf* tools are organized into *fine-grained* and *coarse-grained* categories according to the type of analysis to be carried out. The fine-grained tools aggregate data at cellular level, namely individual cell properties inside regions of interest. These tools are prefixed with 'mqf_fg_'. The coarse-grained *mqf* tools are further separated into '*roi*' and '*grid*' categories to distinguish between regions moving with the tissue and static square regions tiled into a grid. They allow one to visualize and quantify average cell properties at different tissue locations and various spatial scales, and are prefixed with 'mqf_cg_roi_' and 'mqf_cg_grid_' respectively.

To compare fine-grained and coarse-grained cell properties amongst movies we developed a 'multi-db-query' tool, which streamlines the application of the *mqf* tools to a set of movies. To use this tool, the user should first align the movies in time, using convenient morphological or cellular landmarks. As for the *Drosophila* wing, we align movies such that the peaks of cell elongation coincide in the different movies. The user can then apply a chosen *mqf* tool to multiple movies and multiple ROI's. All *mqf* tools, alone or in combination with the 'multi-db-query' tool, generate a table that contains individual or averaged measurements to be visualized on the tissue (*Figure 1 A–E'*, *Figure 2A–D'*, *Figure 3A,D*, *Figure 4B,D*, *Figure 5D*) or in graphs (*Figure 2E–F*, *Figure 3B*, *Figure 4C*, *Figure 5E–F*). This library of tools is described in detail in the TM R-User Manual, which also provides many examples. These tools can be easily extended to address project specific questions.

## Detecting gain and loss of cell contacts

To detect cell neighbor changes, we developed a routine in R that queries the DB and establishes the cell-neighbor relationship at each frame. By comparing the list of neighboring cell identifiers for a given cell between two consecutive frames [$f$, $f + 1$], can one identify and count the changes in neighbor relationships. These can be subdivided into those caused by cell divisions, cell extrusions or the simple gain or loss of a cell contact (not due to division or extrusion). We call these half-T1's because they resemble the gain and loss of cell contacts that occurs during a T1 transition – although they may also be generated by other events such as rosette formation. To assign a neighbor change to the half-T1 category, the corresponding cell identifiers must be present in both frames, ruling out extrusions and cells moving in and out of the field of view. To detect half-T1's that occur simultaneously with divisions, we mask neighbor changes due to divisions by propagating the mother cell identifier (frame $f$) to the two daughter cells (frame $f+1$) that we fuse into one fake cell having the mother cell identifier. We iterate over each pair of consecutive frames and store the half-T1 events due to a gain and a loss of cell neighbors.

## Cell lineages and lineage browsing to follow ROI's forward and backward in time

We pool all lineage information (as contained in the *cell_id*, *left_daughter_cell_id* and *right_daughter_cell_id* columns from the *cell_histories* table) into a directed lineage graph (**Nepusz, 2006**) from which we infer a lineage group identifier and a generation number. By definition the root of each lineage tree is considered as the $F_0$ generation and is thus given a generation value of 0. We follow ROI's backward and forward in time by browsing lineage graphs that were selected based on the regions drawn by using the *draw_n_get_ROIcoord.ijm* Fiji macro. However cells may be lost or not detected when browsing the lineages. One primary reason is that extruding cells are not detected when browsing the lineage backward in time. Cells could also be lost due to possible tracking mistakes. To improve spatial continuity of ROI's we have implemented a method to reassign lost cells to ROI's when located within ROI's. To identify lost cells for a frame within a given ROI, we first distinguish bonds that connect two cells within the ROI, only one cell within the ROI or none. All corresponding cell-pairs define an undirected graph on which a connected component analysis (**Nepusz, 2006**) allows to identify the ROI and non-ROI regions. All cells of non-ROI regions, except for the largest one, are reassigned to become part of the ROI. By doing so, we make the assumption that the largest non-ROI component is defined by the tissue surrounding the ROI.

## Nematic tensors to describe cell elongation and the orientation of cellular processes

When analyzing and visualizing single cell properties, we use the same cell elongation definition as in **Aigouy et al. (2010)**. For a given Cartesian *xy* coordinate system, the elongation of a given cell is defined by the nematic tensor

$$\begin{pmatrix} \epsilon_{xx} & \epsilon_{xy} \\ \epsilon_{xy} & -\epsilon_{xx} \end{pmatrix}$$

with

$$\epsilon_{xx} = \frac{1}{A_c} \int \cos(2\phi) \mathrm{d}A$$

$$\epsilon_{xy} = \frac{1}{A_c} \int \sin(2\phi) \mathrm{d}A.$$

Here, $A_c$ is the area of the given cell, and the integrals are carried out over all points *r* within the cell. The angle is the angle between the vector $r - r_c$ and the *x* axis, where $r_c$ is the cell center defined as

$$r_c = \frac{1}{A_c} \int r \, \mathrm{d}A.$$

Here, the integral is again carried out over all points *r* within the cell. The magnitude of the elongation is given by $\epsilon = \left( \epsilon_{xx}^2 + \epsilon_{xy}^2 \right)^{\frac{1}{2}}$ and the elongation angle $\varphi$ is given by the following two equations

$$\cos(2\varphi) = \frac{\epsilon_{xx}}{\epsilon}$$

$$\sin(2\varphi) = \frac{\epsilon_{xy}}{\epsilon}.$$

Note that this definition of cell elongation is different from the triangle-based definition that is also discussed in this article. However for the fruit fly wing, both cell elongation definitions yield very similar results.

To characterize the axes of cell divisions and T1 transition, we introduce the unit nematic tensors $\tilde{n}_{CD}$, $\tilde{n}_{T1+}$, and $\tilde{n}_{T1-}$. The orientation of a single cell division is quantified by the unit nematic $\tilde{n}_{CD}$ defined by:

$$\tilde{n}_{\mathrm{CD}} = \begin{pmatrix} \cos(2\phi_{\mathrm{CD}}) & \sin(2\phi_{\mathrm{CD}}) \\ \sin(2\phi_{\mathrm{CD}}) & -\cos(2\phi_{\mathrm{CD}}) \end{pmatrix}.$$

Here, the angle $\phi$ is the angle of the line connecting both cell centers with respect to the $x$ axis, measured in counter-clockwise sense. The orientation for a half-T1 transition during which two cell lose neighborship is characterized by:

$$\tilde{n}_{\mathrm{T1+}} = \begin{pmatrix} \cos(2\phi_{\mathrm{T1+}}) & \sin(2\phi_{\mathrm{T1+}}) \\ \sin(2\phi_{\mathrm{T1+}}) & -\cos(2\phi_{\mathrm{T1+}}) \end{pmatrix},$$

where $\phi_{\mathrm{T1+}}$ is the angle of the line connecting the centers of the cells losing neighborship. The orientation for a half-T1 transition during which two cell gain neighborship is characterized by:

$$\tilde{n}_{T1-} = -\begin{pmatrix} \cos(2\phi_{\mathrm{T1-}}) & \sin(2\phi_{\mathrm{T1-}}) \\ \sin(2\phi_{\mathrm{T1-}}) & -\cos(2\phi_{\mathrm{T1-}}) \end{pmatrix},$$

where $\phi_{\mathrm{T1-}}$ is the angle of the line connecting the centers of the cells that gain neighborship. The axes of the nematics $\tilde{n}_{\mathrm{CD}}$, $\tilde{n}_{\mathrm{T1+}}$ and $\tilde{n}_{\mathrm{T1-}}$ roughly correspond to the axis along which the tissue extends due to the respective cell division or half-T1 transition. In particular, because of the minus sign in the definition of $\tilde{n}_{\mathrm{T1-}}$, when the same two cells gain neighborship and lose it again along the same axis, the total effect adding $\tilde{n}_{\mathrm{T1+}}$ and $\tilde{n}_{\mathrm{T1-}}$ is zero.

## Tissue deformation and cellular contributions to it

Here we discuss the formal definitions used to characterize tissue deformation, area change, and shear. We characterize the local rate of tissue deformation by the gradient of the velocity field $v(r)$. We then define the overall deformation rate $V$ of a given piece of tissue by the integral over the area $A_t$ of this piece:

$$V = \frac{1}{A_t} \int \begin{pmatrix} \frac{\partial v_x}{\partial x} & \frac{\partial v_y}{\partial x} \\ \frac{\partial v_x}{\partial y} & \frac{\partial v_y}{\partial y} \end{pmatrix} \mathrm{d}A.$$

This 2x2 tensor can be decomposed into an isotropic part $V^{\mathrm{iso}}$ characterizing the relative growth rate of tissue area, a symmetric, traceless part $\tilde{V}$ characterizing the anisotropic part of the deformation (pure shear rate), and an antisymmetric part $\Omega$ characterizing overall tissue rotation:

$$V = \frac{V^{\mathrm{iso}}I}{2} + \tilde{V} + \Omega e.$$

Here, we have defined $V^{\mathrm{iso}} = \frac{1}{A_t} \int \left( \frac{\partial v_x}{\partial x} + \frac{\partial v_y}{\partial y} \right) \mathrm{d}A$, $\Omega = \frac{1}{2A_t} \int \left( \frac{\partial v_x}{\partial y} - \frac{\partial v_y}{\partial x} \right) \mathrm{d}A$,

$$I = \begin{pmatrix} 1 & 0 \\ 0 & 1 \end{pmatrix}, \quad \tilde{V} = \frac{1}{2A_t} \int \begin{pmatrix} \frac{\partial v_x}{\partial x} - \frac{\partial v_y}{\partial y} & \frac{\partial v_y}{\partial x} + \frac{\partial v_x}{\partial y} \\ \frac{\partial v_y}{\partial x} + \frac{\partial v_x}{\partial y} & \frac{\partial v_y}{\partial y} - \frac{\partial v_x}{\partial x} \end{pmatrix} \mathrm{d}A, \quad \text{and } e = \begin{pmatrix} 0 & -1 \\ 1 & 0 \end{pmatrix}.$$

In recent work, we have shown that the overall shear rate $\tilde{V}$ can be exactly decomposed into a sum of cellular contributions using our Triangle Method (*Merkel, 2014*; *Merkel et al., 2016*):

$$\tilde{V} = \frac{\mathrm{D}\tilde{Q}}{\mathrm{D}t} + T + C + E + D.$$

Here, the nematic tensors $\tilde{Q}$ is the average cell elongation defined based on triangles, and the nematic tensors $T$, $C$, $E$, and $D$ are the shear contributions by T1 transitions, cell divisions, cell extrusions, and correlation effects, respectively. The corotational time derivative $\mathrm{D}\tilde{Q}/\mathrm{D}t$ is defined by

$$\frac{\mathrm{D}\tilde{Q}}{\mathrm{D}t} = \frac{\mathrm{d}\tilde{Q}}{\mathrm{d}t} - 2\left( c\Omega + [1-c]\frac{\mathrm{d}\Phi}{\mathrm{d}t} \right) e \cdot \tilde{Q}.$$

The operator $\mathrm{d}/\mathrm{d}t$ denotes the total derivative, $c = \tanh(2Q)/(2Q)$, and the dot denotes the tensor

dot product. The quantities $Q$ and $\Phi$ denote magnitude and angle of the average cell elongation tensor $\tilde{Q}$.

These formal definitions for $\tilde{Q}$, $\mathrm{D}\tilde{Q}/\mathrm{D}t$, $T$, $C$, $E$, and $D$ refer to deformation rates in the limit of infinitesimal deformations. However, subsequent frames of any real tissue movie are separated by finite time intervals, i.e. finite deformations. There are different ways to adapt these definitions to finite deformations (*Etournay et al., 2015*; *Merkel et al., 2016*). The current implementation of TissueMiner uses the finite-deformation definitions presented in detail in *Etournay et al. (2015)*.

## Acknowledgements

This work was a truly collaborative effort and the authors jointly wrote the manuscript. We are grateful to Christian Dahmann, Marcus Michel and Jacques Boutet de Monvel for critical reading of the manuscript, Benoit Lombardo for his help in Fiji macroing, and Peter Steinbach for useful discussion about workflow engines. We thank Franz Gruber, Vincent Michel and Nathalie Gourreau for testing the quickstart tutorials. RE acknowledges a Marie Curie fellowship from the EU 7th Framework Programme (FP7). This work was supported by the Max Planck Gesellschaft, and by the BMBF. SE acknowledges funding from the ERC.

## Additional information

### Competing interests

FJ: Reviewing editor, *eLife*. SE: Reviewing editor, *eLife*. The other authors declare that no competing interests exist.

### Funding

| Funder | Grant reference number | Author |
|---|---|---|
| European Research Council | | Raphaël Etournay<br>Natalie A Dye<br>Suzanne Eaton |
| Seventh Framework Programme | | Raphaël Etournay |
| Max-Planck-Gesellschaft | | Matthias Merkel<br>Marko Popović<br>Guillaume Salbreux<br>Frank Jülicher |
| Bundesministerium für Bildung und Forschung | | Matthias Merkel<br>Marko Popović<br>Suzanne Eaton<br>Frank Jülicher |
| Alfred P. Sloan Foundation | | Matthias Merkel |
| Gordon and Betty Moore Foundation | | Matthias Merkel |
| National Science Foundation | NSF-DMR-1352184 | Matthias Merkel |

The funders had no role in study design, data collection and interpretation, or the decision to submit the work for publication.

### Author contributions

RE, Participated in regular group discussions to develop the ideas presented, Conception and design, Acquisition of data, Analysis and interpretation of data, Drafting or revising the article; MM, MP, HB, SE, FJ, Participated in regular group discussions to develop the ideas presented, Conception and design, Analysis and interpretation of data, Drafting or revising the article; NAD, Extensively tested TissueMiner and suggested key improvements for TissueMiner, Participated in regular group discussions to develop the ideas presented, Drafting or revising the article, Contributed unpublished essential data or reagents; BA, Developed key image processing and image analysis methods in TissueAnalyzer, Drafting or revising the article, Contributed unpublished essential data

or reagents; GS, Participated in regular group discussions to develop the ideas presented, Conception and design, Analysis and interpretation of data

## Author ORCIDs

Raphaël Etournay, http://orcid.org/0000-0002-2441-9274
Matthias Merkel, http://orcid.org/0000-0001-9118-1270
Marko Popović, http://orcid.org/0000-0003-2360-3982
Guillaume Salbreux, http://orcid.org/0000-0001-7041-1292
Frank Jülicher, http://orcid.org/0000-0003-4731-9185

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

## Appendix 1

### Parsing tracked-cell images to build the TM-DB

We used TissueAnalyzer to detect cell contours (segmentation) and to track cells over time. This software generates two output masks - the *tracked-cell* and the *cell-division* masks. These masks are raster images. In both masks, cell circumferences are represented by one pixel thick white lines. In the *tracked-cell* mask, all pixels inside the cell circumference have the same unique color. In consecutive frames, the same cell has the same color. In the *cell-division* mask, each cell is colored either in black or in blue. If a cell is blue, it is a daughter cell that emerged from a division between two consecutive frames. Otherwise, a cell is black in the *cell-division* mask.

We wrote a custom C++ parser that converts information contained in the *tracked-cell* and *cell-division* masks into tables that can be easily transformed into the TissueMiner database. This parser first extracts topological and geometrical information about cells, bonds and vertices for each individual frame. Afterwards, it analyzes the continuity of cell existences across consecutive frames. In particular, it tries to infer reasons for appearance or disappearance of cells. Finally, based on this information, history and lineage can be established for each cell (see Materials and methods).

The parser extracts the topological information for each frame from the *tracked-cell* mask. It scans the entire mask image row by row. Whenever it hits a cell boundary (white pixel), it defines the cell circumference and divides it into bonds defined as contiguous white pixels that are in contact with exactly two cells, and vertices defined as white pixel surrounded by 3 or more pixels of different colors). The topology, namely how neighboring cells are arranged around each cell, is obtained by creating a counter-clockwise series of consecutive directed bonds. Each directed bond stems from a unique vertex and points to the next vertex along the cell circumference. We created the concept of directed bonds to unambiguously characterize the wiring between cells, vertices, and (undirected) bonds (*Figure 6D*). The parser stores the topology by creating the relation of each directed bond with its next counter-clockwise follower on the cell circumference and with the vertex from which it stems (*Figure 6D*). To store the cell-nearest-neighbor relationships, we map each cell-cell contact (bond) to the two corresponding directed bonds, where each directed bond is associated with a single cell and a single vertex. This is illustrated in *Figure 6D*, where the cyan directed bond points towards vertex $i$ and lies on the side of cell $\alpha$, whereas the magenta directed bond points towards vertex $j$ and lies on the side of cell $\beta$. We call the cyan and magenta directed bonds to be 'conjugated' to each other.

The parser also extracts geometrical information for each given cell by going along the circumference of that cell. Cell area $A$ is computed as:

$$A = \frac{1}{2}\sum_i \left[ p_x(i)p_y(i+1) - p_y(i)p_x(i+1) \right],$$

where the index $i$ runs over all pixels in counter-clockwise order around the cell. The vector $\vec{p}(i) = \left( p_x(i), p_y(i) \right)$ denotes the position of pixel $i$. The cell center $\vec{c}$ is computed as:

$$\vec{c} = \frac{1}{6A}\sum_i \left[ p_x(i)p_y(i+1) - p_y(i)p_x(i+1) \right]\left[ \vec{p}(i) + \vec{p}(i+1) \right]$$

Cell shape anisotropy is described by the two components of the symmetric traceless tensor defined elsewhere (*Aigouy et al., 2010*). The cell perimeter is computed as the sum of the lengths of all bonds belonging to the cell boundary. The length of a bond is computed as the summed pixel distance going along this bond pixel by pixel. In particular, when advancing on

pixel up, down, left, or right, one is added to the bond length. However, when advancing diagonally, $\sqrt{2}$ is added.

After extracting topology and geometry for each frame, the parser infers for each cell whether it stays in the tissue, or whether it appears or disappears in going from one frame $f$ to the next one $f + 1$. Which of the three possibilities occurs can be directly inferred using fact that each cell is assigned a unique color throughout all *tracked-cell* masks. If a cell is present in both *tracked-cell* masks, it is just staying within the tissue. If it is only present in frame $f + 1$, it is appearing, which may happen for several reasons. For one, a cell may appear as a daughter cell of a division, which can be checked using the *cell-division* mask. Moreover, if a cell appears at the margin of the piece of tissue, it is declared as moving in via the margin. The same happens if an appearing cell is next to a cell that has already been declared as moving in via the margin. If none of these happened, the parser declares a segmentation error ('SegErrAppearance') as the reason for appearance.

If a cell is only present in frame $f$ but not in $f + 1$, it is disappearing, which may happen for several reasons, too. For one, the cell could be the mother cell of a division that occurs between frames $f$ and $f + 1$. This can be checked using the *cell-division* mask. Otherwise and if the cell is disappearing at the margin, the parser marks the cell as moving out of the margin. The same happens if the disappearing cell is next to a cell that has already been marked as moving out of the margin. Finally, every cell that disappears for none of the two previous reasons is marked as undergoing an extrusion/apoptosis.

## Implementing the TissueMiner relational database

The TissueMiner parser generates tables from which we build the TissueMiner relational database (TM-DB). To do so, we used the formalism developed in the Merise method (*Tardieu et al., 2000*), which includes the entity-relationship model (*Peter Pin-Shan, 1976*), the relational database theory (*Codd, 1970*, *1972*) and Codd's normal forms (*Codd, 1971*, *1974*); thus, it allows one to translate the conceptual data model into a relational database scheme.

We first establish the 'entity-relationship' scheme of the database to represent the information extracted with the parser in entities, and to establish relationships between and within entities. This conceptual approach defines the basic elements of the entity-relationship model (*Peter Pin-Shan, 1976*): the entity, the association, the cardinality and the identifier. Entities consist of objects (*cells*, *bonds*, *vertices*, *frames*) or concepts (*cell_histories*, *directed_bonds*) that can be uniquely identified. The association is a link that relates two entities. The identifier is an obligatory property of an entity and uniquely defines each occurrence of the entity. The cardinality reflects the minimum and maximum connections (functional dependencies) between the identifiers of two associated entities: [1,n] stands for one-to-many, [0,n] for none-to-many, [1,1] for one-to-one, and [0,1] for none-to-one. Hence, each association is assigned two cardinalities corresponding to the 2 possible directions of association between the two entities. For the sake of clarity, *Figure 6A* shows a simplified 'entity-relationship' scheme of the TM-DB without cardinalities. However, cardinalities are used in the Merise method to translate the conceptual scheme (*Figure 6A*) into the logical scheme shown in *Figure 6—figure supplement 2A*. We therefore show them in *Table 1*. The rules to translate a conceptual scheme to a logical one can be found here (*Tardieu et al., 2000*). Below, we explain our conceptual scheme along with its translation into the logical scheme, which can be directly implemented using a chosen SQL language. Applying these rules to our TM-DB, these entities become physical tables in the logical scheme, and associations become table columns ('foreign keys' in blue) in related tables (*Figure 6—figure supplement 2A*). The foreign keys constitute a referential integrity constraint between tables.

The TM-DB consists of six entities, *frames, cells, vertices, bonds, directed_bonds* and *cell_histories* that are linked by logical associations (*Figure 6A*). Their respective identifier is

underlined in the conceptual scheme (*Figure 6A*), and becomes the 'primary key' placed in the table header in the logical scheme (*Figure 6—figure supplement 2A*). In the TM-DB, identifiers (*frame*, *cell_id*, *vertex_id*, *bond_id*, *dbond_id*) are numbers that we use to index the corresponding tables. Time and movie frames are contained in the *frames* entity. Geometrical information is contained in the *cells*, *vertices* and *bonds* entities. Topological information including cell neighbor relationships is represented in the *directed_bonds* entity. The cell ancestry is represented in the *cell_histories* entity.

**Table 1.** Cardinalities per association.

| Entity A | Entity B | Association (A->B) | Cardinality A->B | Cardinality B->A |
|---|---|---|---|---|
| *cell_histories* | *cell_histories* | to be daughter of | [0,1] | [0,n] |
| *cells* | *cell_histories* | to belong to | [1,1] | [1,n] |
| *cells* | *frames* | to exist in | [1,1] | [1,n] |
| *directed_bonds* | *cells* | to be part of | [1,1] | [1,n] |
| *directed_bonds* | *directed_bonds* | to be conjugated | [1,1] | [1,1] |
| *directed_bonds* | *directed_bonds* | to be next left | [1,1] | [1,1] |
| *directed_bonds* | *frames* | to exist in | [1,1] | [1,n] |
| *directed_bonds* | *bonds* | to be part of | [1,1] | [1,n] |
| *directed_bonds* | *vertices* | to stem from | [1,1] | [1,n] |
| *vertices* | *frames* | to exist in | [1,1] | [1,n] |
| *bonds* | *frames* | to exist in | [1,1] | [1,n] |

In order to relate a given cell to its lineage and intrinsic properties during the time evolution of the movie, we create specific associations within and between the *cells* and *cell_histories* entities. In the *cell_history* entity, a cell is uniquely determined by a cell identifier (*cell_id*) that exists as long as the tracked cell does not die or divide. All cells are represented in this entity, which stores in which frame a given cell appears (*first_occ*) and disappears (*last_occ*), and why (*appears_by* and *disappears_by*). The cell ancestry is represented by the 'be_daughter_of' association that relates each dividing cell to its two daughters (*left_daughter_cell_id* and *right_daughter_cell_id* columns, *Figure 6—figure supplement 2A*). To relate a cell to the time evolution of its properties (center of mass, area, shape anisotropy, polarized protein distribution), we create an association between the *cells* and *cell_histories* entities, in which each entry is uniquely determined by the combination of *cell_id* and *frame*. As movies may be acquired at different frame rates, we also represent the real time evolution (in seconds) in the *frames* entity that we connect to the *cells* entity.

To represent the cell topology in the database, we create a *directed_bond* entity along with a self-association 'be next left' that links each directed bond in each frame (*dbond_id*) to its next counter-clockwise follower (*left_dbond_id* column, *Figure 6—figure supplement 2A*). This stores the ordering of the directed bonds around each cell. To relate each cell with its neighbors in each frame, we define a 'be conjugated' self-association that links each directed bond to its corresponding conjugated bond (*conj_dbond_id* column, *Figure 6—figure supplement 2A*). To connect the topology to geometrical information, we first define an additional association ('be part of') that connects the *cells* to the *directed bonds* entities. We then connect both entities to the *frames* entity by defining the association 'exist in' that matching the *frame* attribute (*Figure 6—figure supplement 2A*). Finally, we connect *directed bonds* to *bonds* and *directed bonds* to *vertices* by creating the associations 'be part of' and 'stem from', respectively (see *vertex_id* and *bond_id* columns, *Figure 6—figure supplement 2A*).

The TM-DB follows the 3 first normal forms established by *Codd (1971, 1974)*. The first normal form ensures that all entity properties are mono-valued and non-divisible, and that at least one of them is the identifier, which semantically determines all other properties of the

entity. The second normal form adds constraints on the identifiers: if an identifier is composed of multiple properties (see *cells* entity), the other properties must be determined by the whole identifier and not by only part of it. The third normal form stipulates that a property isn't allowed to be determined by an existing property that isn't an identifier. In the conceptual scheme, those 3 normal forms ensure that the identifier uniquely defines each property of the entity. They also ensure that entity properties are entirely determined by the sole identifier. This helps clarifying the notion of entities and their content when creating the data model. It also helps reducing redundancy in the database.

The logical scheme of the TM-DB is implemented using the SQLite management system (*Jay, 2010*). We chose SQLite for its ease of use: there is no need to install a dedicated server and the DB is stored in a single file that is easily shared with collaborators. The source code is accessible on GitHub repository (see *Box 1*).

