## [Decision Letter]

Thank you for submitting your work entitled "TissueMiner: a multiscale analysis toolkit to quantify how cellular processes create tissue dynamics." for consideration by *eLife*. Your article has been reviewed by two peer reviewers, and the evaluation has been overseen by a Reviewing Editor and Naama Barkai as the Senior Editor.

The reviewers have discussed the reviews with one another and the Reviewing Editor has drafted this decision to help you prepare a revised submission.

Summary:

Large scale tissue imaging is becoming an everyday tool in developmental biology, but high throughput analysis methods are restricted to few laboratories. Etournay and co-workers present a software platform for segmentation and tracking of 2D epithelial monolayers. The software provides a number of geometric outputs including deformation tensors and cell connectivity. As the tools within TissueMiner appear to be identical to those used by the authors previously (2015 *eLife* publication), the value of TissueMiner lies not its novelty or proof-of-concept, but squarely in the likelihood of it being used by an average researcher. Both reviewers felt that the software and current tutorials, while expansive, were far from user-friendly in their current implementation, and the tutorials, while extensive, assumed far more technical background than the average user would have. Nevertheless, both reviewers felt that TissueMiner, if improved as described below, would be an important advance and could be very useful to the community.

Some of the key problems encountered were:

1) Installation of the software is not user-friendly. Between the large size of the sample datasets (2+ GB each), and tedious GitHub interactions presented as a series of code fragments rather than a single script, it took one reviewer over an hour to get the sample data downloaded, and to deal with various docker virtual machine bugs. Even pulling the demo data down caused bugs due to permission errors that were not addressed in the tutorial.

2) Once docker was running, execution of the initial data processing step resulted in a dizzying array of analysis steps with no explanation of what was happening or how much time it would take.

3) Users must already have R installed, but this is not mentioned.

4) Preparing TissueMiner to run through RStudio should not require the user to copy-paste code fragments each time. A single script that does this should be included.

5) Tissue Analyzer – the preprocessing tool made by the same group and essentially required for use with Tissue Miner – works beautifully and has a series of video tutorials. This would be a good template for TissueMiner.

6) RStudio initialization script kept crashing. Apparently additional toolboxes are required, such as devtools.

Essential revisions:

In the context of the software, the reviewers specifically recommended that you:

1) Make the sample data much smaller in size (<2GB) so that the initial docker processing takes <5 min, or at least has a remaining time clock;

2) Make the software completely scripted, including all the initial importing in RStudio (no copy-pasting, just loading an RStudio file);

3) Prepare a new "First Use" R tutorial, skipping most of the R-101 training and just getting a user up and running with something simple, like plotting cell-area.

Then:

1) Conduct user testing with testers who truly have "no programming experience". Allow them to test installation *unsupervised*.

2) Create more streamlined, focused tutorials rather than an enormous master guide. TissueMiner tries to do everything at one time (e.g. the initial docker processing), and it is difficult to follow. Give the user the simplest data-set you can provide, and have them perform the simplest analysis (e.g. only focus on cell area, or tracking, or divisions).

3) The authors illustrate the capabilities of the software using the case of wing vein morphogenesis, but this is not a validation. To validate the software, the authors could use computer generated images of cell monolayers with known geometry, and demonstrate that the software is able to recover the different geometric parameters of each cell in the tissue.

---

## [Author Response]

Some of the key problems encountered were:

1) Installation of the software is not user-friendly. Between the large size of the sample datasets (2+ GB each), and tedious GitHub interactions presented as a series of code fragments rather than a single script, it took one reviewer over an hour to get the sample data downloaded, and to deal with various docker virtual machine bugs. Even pulling the demo data down caused bugs due to permission errors that were not addressed in the tutorial.

To address these problems, we have significantly simplified the installation procedure. For Ubuntu Linux, we now provide a single script *install_tm.sh* that fully installs TissueMiner. Instructions are found directly on the landing page of the TissueMiner GitHub repository: https://github.com/mpicbg-scicomp/tissue_miner#installation. For other systems, we provide a 2-step protocol for installing TissueMiner: first install the Docker Engine and second download the *TissueMiner docker image*. The installation procedure of this dockerized TissueMiner is explained in detail on the TissueMiner GitHub site: https://github.com/mpicbg-scicomp/tissue_miner#installation

The new installation procedure was successfully tested by two users with no programming experience on Ubuntu 14.04, Ubuntu 15.10, and on different versions of MacOSX (Mavericks, Yosemite and El Capitan).

We enhanced the former “R tutorial” and renamed it “TM R-User Manual”: https://mpicbg-scicomp.github.io/tissue_miner/user_manual/TM_R-UserManual.html

In addition, we now provide simplified “Quickstart Tutorials” that are designed to help new users quickly learn the procedures and capabilities of TissueMiner:

https://github.com/mpicbg-scicomp/tissue_miner#documentation

TM R-User Manual and Quickstart Tutorials are now based on a demo dataset that is smaller than that which was originally provided (~35Mb of compressed data). Nonetheless, we also provide the large datasets for more sophisticated analysis or for those interested in pupal wing morphogenesis. This larger dataset is now compressed (<800Mb each) to facilitate download and is used in the last part of the documentation (see TM R-User Manual, section 3).

2) Once docker was running, execution of the initial data processing step resulted in a dizzying array of analysis steps with no explanation of what was happening or how much time it would take.

To improve clarity and speed, we streamlined the initial processing (removing many of the automatic video creations) and now only display messages that indicate the progression of the workflow. With these changes, the initial data processing steps in the automated workflow require less than 3min on the provided demo sample.

The new Quickstart Tutorials are designed to guide the user through the database creation as well as through specific type of analysis (cell area, elongation, etc…) by running appropriate R scripts (a few seconds of run time for each script using the demo dataset). Their source code can be found on the GitHub repository:

https://github.com/mpicbg-scicomp/tissue_miner/tree/master/docs/quickstart/scripts

3) Users must already have R installed, but this is not mentioned.

On Ubuntu, our new installer script *install_tm.sh* solves this issue. On other systems, we provide a docker image of TissueMiner that doesn’t require additional R installation. Details about how to configure and use this dockerized TissueMiner for developing custom R scripts can be found in the TM R-User Manual (section 1.3.1):

https://mpicbg-scicomp.github.io/tissue_miner/user_manual/TM_R-UserManual.html

Optionally, MacOSX users may also find easier to install R, Rstudio and the TissueMiner API independently from the docker image (just a matter of taste). We therefore provide an installer *install_tm_api.sh*. Instructions are present in the TM R-User Manual(section 1.3.1).

4) Preparing TissueMiner to run through RStudio should not require the user to copy-paste code fragments each time. A single script that does this should be included.

We now provide a single script entitled analyze_movie.R that combines all individual quickstart tutorial scripts. This script is present on the GitHub repository: https://github.com/mpicbg-scicomp/tissue_miner/tree/master/docs/quickstart/scripts

Instructions about how to run this script can be found in the Quickstart R-Tutorials.

Additional explanations and code fragments for advanced users of TM have been moved to the TM R-User Manual.

5) Tissue Analyzer – the preprocessing tool made by the same group and essentially required for use with Tissue Miner – works beautifully and has a series of video tutorials. This would be a good template for TissueMiner.

Because TissueMiner is based on command line input, rather than graphical input as in TissueAnalyzer, we believe video tutorials are not as useful. Instead we designed the web-based tutorials: Quickstart TM-R Tutorials for Ubuntu and Quickstart docker-TM-R Tutorials for other systems. These tutorials guide even complete novices through the use of TissueMiner by providing not just instructions but commands that can be directly copy-pasted into the terminal to run the analyses:

https://github.com/mpicbg-scicomp/tissue_miner#documentation

We provide video tutorials for the ImageJ macros in TissueMiner (for drawing regions of interest and orienting the tissue).

*6) RStudio initialization script kept crashing. Apparently additional toolboxes are required, such as devtools.*

As mentioned in point 3), the new installation procedure resolves this problem.

Essential revisions:

In the context of the software, the reviewers specifically recommended that you:

1) Make the sample data much smaller in size (<2GB) so that the initial docker processing takes <5 min, or at least has a remaining time clock;

As explained above, we now provide a smaller sample dataset (a cropped region of the pupal wing, ~35Mb in size) to guide new users through the Quickstart Tutorials. Furthermore, the user can now select specific steps of the analysis to run. For example, it may not be necessary or desired at the beginning of the tutorial to create videos of patterned cell behaviors. For this example dataset, a simplified initial analysis workflow, including database creation, cell neighbor change detection and shear calculation, requires less than 3 min on a single core(2.7 GHz Intel Core i5; 2Gb of RAM). Subsequent analysis steps, such as video creation, can be run on demand by selecting the corresponding *snakemake* rules, see Figure 7 and TM R-User Manual.

The R programming language doesn’t easily allow one to systematically provide a progress bar. Nonetheless, we estimated the processing time required for three datasets of increasing size (excluding extra video creation):

**Table d36e6345:** 

dataset	Compressed Size (Mb)	True size (Gb)	Number of cell-cell contacts	Number of cell contours	Number of cell lineages	Number of ROIs	Run time
demo	31	0.1	~200 000	~68 000	~1200	3	2min48sec
WT_3	563	2.1	~1 450 000	~487 000	~8600	6	17min00sec
WT_1	786	2.5	~1 610 000	~540 000	~9400	6	18min03sec

The size of WT_2 dataset is similar to the size of WT_1.

Running these analyses on multicore computers would significantly reduce these times.

2) Make the software completely scripted, including all the initial importing in RStudio (no copy-pasting, just loading an RStudio file);

The full analysis of single movies is now completely encoded in a single script, *analyze_movie.R*. This script is explained in the Quickstart Tutorials and can either be used as is or custom edited in Rstudio.

Comparisons between multiple movies require adaption to the specific tissue being analyzed. Therefore, we provide a single template script, *compare_multiple_movies.R*, which requires the user to enter the location of the movies and/or ROIs to compare.

*3) Prepare a new "First Use" R tutorial, skipping most of the R-101 training and just getting a user up and running with something simple, like plotting cell-area.*

As requested, we developed “First Use” Quickstart Tutorials that quickly introduces novices to the capabilities and procedures of TissueMiner.

To help users that are interested in further developing TissueMiner in R for more sophisticated and customized analyses, we still provide some information about R, including the grammar of data manipulation and graphics, in a dedicated tutorial referenced in the TM R-User Manual: https://mpicbg-scicomp.github.io/tissue_miner/user_manual/Learning_the_R_basics_for_TissueMiner.html

Then:

1) Conduct user testing with testers who truly have "no programming experience". Allow them to test installation unsupervised.

Two testers with "no programming experience" managed to install and run the Quickstart Tutorials on both Ubuntu and MacOSX using the updated installation procedures.

We also provide two additional tutorials to help users getting started with the analysis of their own data. For Ubuntu:

https://github.com/mpicbg-scicomp/tissue_miner/blob/gh-pages/quickstart_tutorial/ubuntu/tm_qs_user_data.md#first-use-of-tissueminer-with-your-own-data

For other systems:

https://github.com/mpicbg-scicomp/tissue_miner/blob/gh-pages/quickstart_tutorial/other_os/tm_qs_user_data.md#first-use-of-tissueminer-with-your-own-data

2) Create more streamlined, focused tutorials rather than an enormous master guide. TissueMiner tries to do everything at one time (e.g. the initial docker processing), and it is difficult to follow. Give the user the simplest data-set you can provide, and have them perform the simplest analysis (e.g. only focus on cell area, or tracking, or divisions).

The new Quickstart Tutorials were designed to address this point. We also created the TM R-User Manual in html format, including a table of contents and links to each type of analysis, for those with more experience with R.

3) The authors illustrate the capabilities of the software using the case of wing vein morphogenesis, but this is not a validation. To validate the software, the authors could use computer generated images of cell monolayers with known geometry, and demonstrate that the software is able to recover the different geometric parameters of each cell in the tissue.

We produced two computer-generated movies of tissues with known geometry and deformation. One dataset corresponds to hexagonal cells undergoing homogeneous isotropic expansion with an imposed rate, and the other corresponds to hexagonal cells undergoing homogeneous pure shear with an imposed rate. We segmented and tracked both movies and performed the entire TissueMiner deformation analysis. We found that TissueMiner correctly recapitulates the imposed deformations and their cellular contributions. Moreover, we find for these movies that expected deviations due to pixelation are two orders of magnitude smaller than the deformations that we aim to measure.

We added a subsection entitled “Validation of tissue deformation measurements using computer-generated cells” – where we discuss all of these points.